# Task-induced attention load guides and gates unconscious semantic interference

Shao-Min Hung 🔘 [1,2✉], Daw-An Wu 🔘 [1] & Shinsuke Shimojo 🔘 [1,3]

The tight relationship between attention and conscious perception has been extensively researched in the past decades. However, whether attentional modulation extended to unconscious processes remained largely unknown, particularly when it came to abstract and high-level processing. Here we use a double Stroop paradigm to demonstrate that attention load gates unconscious semantic processing. We find that word and color incongruencies between a subliminal prime and a supraliminal target cause slower responses to non-Stroop target words—but only if the task is to name the target word (low-load task), and not if the task is to name the target's color (high-load task). The task load hypothesis is confirmed by showing that the word-induced incongruence effect can be detected in the color-naming task, but only in the late, practiced trials. We further replicate this task-induced attentional modulation phenomenon in separate experiments with colorless words (word-only) and words with semantic relationship but no orthographic similarities (semantics-only).

[1] Biology and Biological Engineering, California Institute of Technology, 1200 E. California Blvd., Pasadena, CA 91125, USA. [2] Huntington Medical Research Institutes, 686 South Fair Oaks Avenue, Pasadena, CA 91105, USA. [3] Computation and Neural Systems, California Institute of Technology, 1200 E. California Blvd., Pasadena, CA 91125, USA. ✉email: smhung@caltech.edu

The role of attention in modulating perceptual processes has been a long-standing topic in Psychology and Neuroscience. As attention operates on distinct stimulus/perceptual/visual properties, attention can be described as a collection of specialized mechanisms operating on the basis of stimulus properties such as space, time, features, etc. (see a review[1]). On the other hand, the load theory of attention describes attention as a more general processing resource that spans across specialized perceptual mechanisms and considers how the availability of such fluid resources could gate perceptual processing[2,3]. Whether attention is treated as something which is directed to a specific entity or as something utilized as general cognitive resources, studies have established the benefits of attention in various tasks over the past decades. For instance, when an identical feature (e.g. motion direction) is shared between two sets of moving dots, behavioral performance on comparing the speed of the two improves[4]. This is interpreted as feature-based attention enhancing performance on the feature-sharing stimuli. Similarly, orienting attention to a specific location improves visual performance[5] and even boosts stimulus contrast[6]. These findings suggest that attentional modulation on visual information begins very early in the visual pathway and affects rudimentary visual features. Indeed, neurophysiological evidence has shown that attentional modulation begins early both anatomically (primary visual cortex (V1)[7]; lateral geniculate nucleus (LGN)[8,9]) and temporally (C1/P1 ERP component[10,11]: 100 ms after stimulus onset).

Attention not only benefits the processing of conscious stimuli, it has also been shown that attention operates near the border of consciousness. In an example of feature-based attention extending to the processing of near-threshold stimuli, Rossi and Paradiso[12] found that observers who were focused on a foveal Gabor patch would detect a near-threshold peripheral grating at higher rates if the two had similar orientations and spatial frequencies. In an experiment treating attention as a fluid resource, Cartwright-Finch and Lavie[13] showed that a peripheral distractor was more likely to enter conscious awareness and be detected when the observer was under a condition of low perceptual load, compared to a high-perceptual-load condition. Finally, in the phenomenon of inattentional blindness, focusing attention on a specific aspect of a visual scene prevents the conscious awareness of salient stimuli, even if they are presented foveally[14,15]. These studies have established that attention operates at the boundary of consciousness and may serve as a gating system determining whether visual stimuli enter into conscious awareness. Clearly, attention is tightly intertwined with consciousness both from our subjective experiences and experimental findings.

The next question is whether attention operates on unconscious visual content. Studies addressing this question provide a critical window for examining the relationship between the two. Are attention and consciousness one and the same, or can attention still modulate our visual system in the absence of conscious awareness? More and more findings suggest attentional modulation on subliminal stimuli. For example, both Bahrami et al.[16] and Kanai et al.[17] showed orientation adaptation from interocularly suppressed subliminal orientation. As in the previous examples, the unconscious effect was modulated by either feature-based attention elicited from attending a visible orientation[17] or the perceptual load induced by a concurrent central task[16]. Further, Bahrami et al.[16] showed that the availability of attention strengthened primary visual cortex signal for a subliminal distractor. Finally, Hsieh et al.[18] showed that subliminal singletons elicited a location-specific cuing effect, prompting higher accuracy on a subsequent visual task. The effect disappeared when participants were instructed to perform a demanding concurrent task. These studies together suggest that deployment of attention to the unconscious stimuli enhances unconscious processes while depletion of attention drains the resources necessary for unconscious processes. That is, (1) under a high-load condition or (2) when attention is directed away from unconscious stimuli, the effect from such stimuli either weakens or disappears.

However, past studies showing that attention operates on subliminal visual content have been constrained to low-level visual features such as simple orientation and stimulus saliency (low-level visual processing refers to light-based, retinotopic, early stages simple feature processing, such as contrast, orientation and color (e.g.[19]), while high-level visual processing refers to categorical or semantic extraction of visual input, which is relatively invariant to the viewing conditions such as luminance and angles). These past studies thus suggest that, unlike conscious processes, the attentional effect on subliminal stimuli may be confined in the early visual cortex and to simple low-level features, leading to a dichotomy of attentional modulation on conscious and unconscious stimuli. On the other hand, it remains possible that attention may modulate subliminal visual content from early to late stages, leading to attentional modulation on a wide spectrum of subliminal visual information. That is to say, whether attentional modulation operates on the high-level processing of subliminal stimuli remains largely unknown. The interaction between attention and subliminal low-level/high-level features also tackles an important question: the robustness of unconscious effects in the scientific literature. The existence of certain high-level unconscious processes has been shown sporadically, often without stable replication. One possibility would be that the strength of subliminal high-level visual information is inherently weaker, and more attentional resources have to be directed to such information in order to maintain a stable representation and to give rise to an experimental effect.

The current study thus set out to directly examine whether the availability of attentional resources plays a key role in modulating subliminal high-level visual information. We introduced a double Stroop paradigm in which one Stroop word was interocularly suppressed and served as a subliminal prime and a subsequent Stroop word was presented as a supraliminal target (Fig. 1). Participants were instructed to name the word or color of the supraliminal Stroop word. In a typical Stroop paradigm, the participant is presented with a colored word (e.g. the word "BLUE" printed in red) and asked to name either its color or the word. Color-naming suffers from strong interference from the semantics of the word, which is relatively automatically processed and difficult to surpress (Stroop effect). On the other hand, word-naming suffers only weak interference from the processing of word color (reverse Stroop effect). We took advantage of this asymmetry of task load in responding to different aspects of the stimulus and defined color-naming as high-load and word-naming as low-load in our study[20]. While keeping the procedure and stimuli identical across different experimental conditions, the word-naming and color-naming tasks served to selectively engage participants on one aspect of target feature and modulate the task load. We measured the effects of semantic and color incongruency between a subliminal and a supraliminal Stroop word, using reaction time as a gauge of that high-level visual subliminal content. We then examined how the attentional modulation generated by conscious task demands could spread to that subliminal processing effect. The level of task load could be one influence on subliminal processing. Further, responding to different aspects of a target word could selectively activate feature-specific attention, which could be further utilized to select the subliminal relevant feature. That is, responding to the semantic feature of a supraliminal word could selectively deploy semantic-specific attention to the subliminal semantic feature.

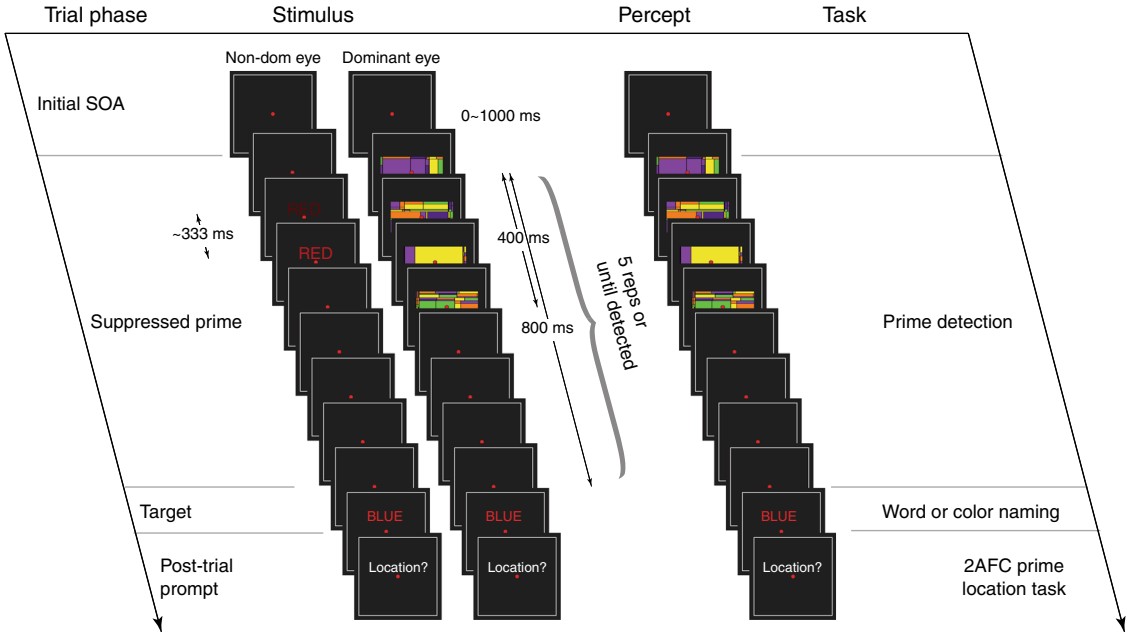

**Fig. 1 Trial sequence, stimulus, and task.** Each trial was self-paced and began with a varied stimulus-onset asynchrony (SOA) ranging from 0.1–1s. After which a dynamic flashing colored Mondrian pattern was presented to the dominant eye while the colored word prime was presented to the non-dominant eye. During 400 ms suppression period, the suppressed word was sandwiched by the Mondrian pattern by two frames at each end, leading to 333 ms presence. The 400-ms-on-400-ms-off pattern was repeated five times or until participants reported breakthrough. If breakthrough was reported, the trial ended immediately. If not, another colored word was presented immediately until response. Participants were instructed to name the word (Experiments 1, 3, and 5, 7) or color (Experiments 2, 4, 6, 8) of the target. A 2-alternative-force-choice location task was present at the end of each trial, participants were instructed to report the location of the suppressed prime. While the prime detection served as a subjective report of prime visibility, this location task served as a post-trial objective gauge of prime visibility. The prime was occasionally superimposed on the Mondrians (visible catch) or simply non-existent (blank catch). Refer to Table 2 for detailed prime-target combinations in all experiments. The prime and target were of different font sizes and presented on slightly jittered locations.

**Table 1 Objective measures on behavioral performance.**

| Experiment | Visible catch detection ACC (%) | Blank catch detection ACC (%) | Prime location task ACC (%) | Prime breakthrough rate (%) |
|---|---|---|---|---|
| 1 | 97.19 (1.15) | 94.06 (1.40) | 48.86 (2.17) | 23.42 (1.96) |
| 2 | 98.44 (0.77) | 97.81 (1.04) | 50.07 (1.69) | 24.09 (1.62) |
| 3 | 99.38 (0.43) | 97.16 (0.85) | 51.08 (1.32) | 28.75 (3.65) |
| 4 | 98.75 (0.97) | 97.26 (1.59) | 51.80 (1.27) | 22.47 (0.33) |
| 5 | 97.81 (1.14) | 98.98 (0.24) | 51.03 (1.06) | 25.75 (2.25) |
| 6 | 98.75 (0.73) | 98.05 (0.78) | 51.00 (1.28) | 23.28 (1.43) |
| 7 | 98.12 (1.29) | 98.35 (0.53) | 49.81 (1.15) | 22.78 (0.42) |
| 8 | 99.69 (0.31) | 98.45 (0.50) | 50.20 (1.33) | 22.81 (0.23) |

Participants' performance on two types of catch trials (visible catch and blank catch), suppressed prime localization, and prime breakthrough rate across all experiments. These figures are drawn and presented in Supplementary Fig. 1.

Two non-exclusive hypotheses can thus be raised and examined. (1) An unconscious effect from high-level visual content requires general attentional resources. That is, an unconscious effect will occur only when the task load is low and additional attentional resources could be distributed to the unconscious stimulus. (2) An unconscious effect from high-level visual content requires feature-specific attention. That is, an unconscious effect will occur only when the task set is tuned to the critical subliminal feature.

## Results

**Conscious task demands gated unconscious processes**. To first establish that the suppressed prime was invisible, we examined the accuracy of the location task in Experiment 1 (word-naming). The mean accuracy was 48.86% (2.17%) and not different from chance (paired $t(19) = -0.53$, $p = 0.61$), indicating that the suppression

was successful. Moreover, the performance on the blank and visible trials was 94.06% (1.40%) and 97.19% (1.15%), respectively, showing that the participants responded with high accuracy and consistency. A similar pattern was found in Experiment 2. The mean accuracy on the location task was 50.07% (1.69%) and not different from chance rate (paired $t(19) = 0.04$, $p = 0.97$). Moreover, the performance on the blank and visible trials on the detection task was 97.81% (1.04%) and 98.44% (0.77%), respectively, showing high accuracy and consistency. Table 1 summarizes these objective measures in all experiments.

To examine whether prime-target word and color congruency affected target responses, we put in three different factors into our analysis: prime-target color congruency, prime-target word congruency, and target word-color congruency (that is, whether target is a Stroop word (word-color incongruent) or a non-Stroop word

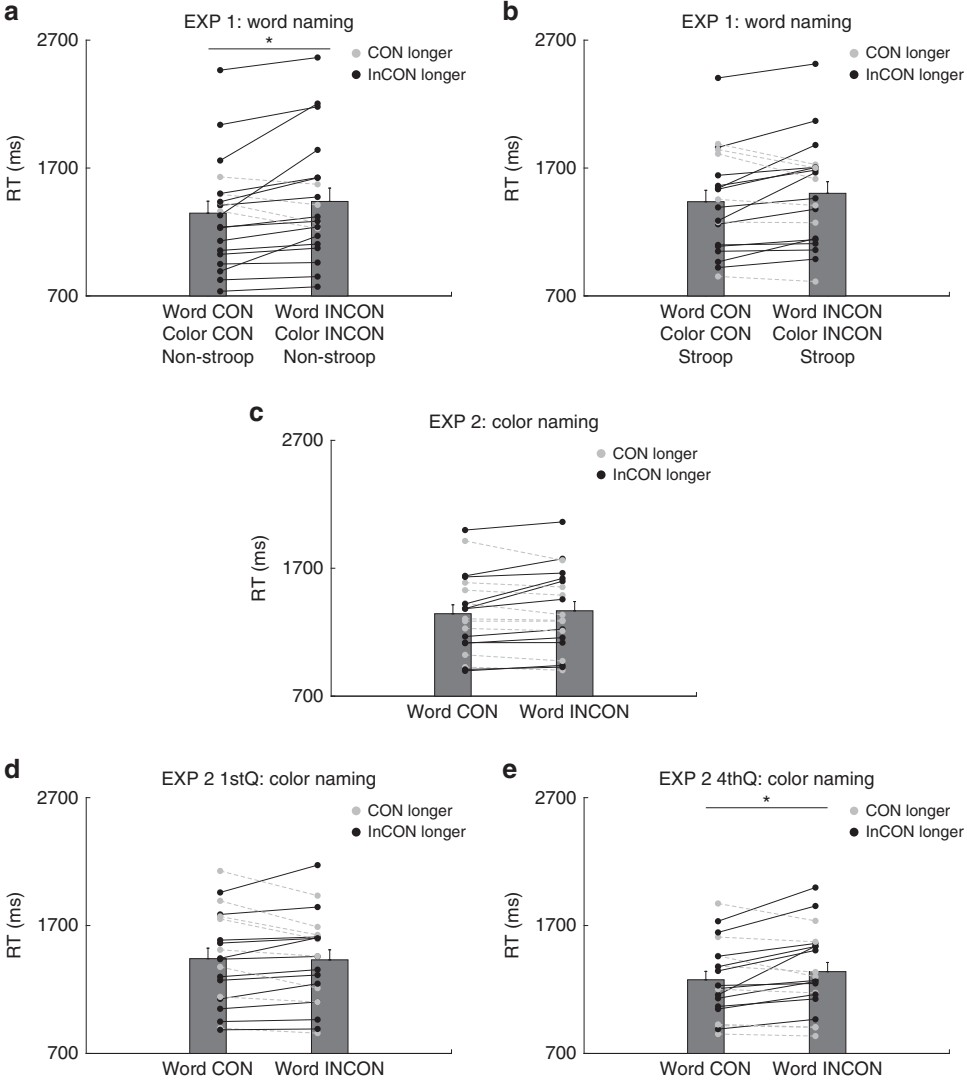

**Fig. 2 Experiments 1 and 2 results.** $n = 20$ in each experiment. Y-axis denotes reaction time, and X-axis denotes congruent and incongruent conditions between the subliminal prime and supraliminal target (CON: congruent; INCON: incongruent; Non-Stroop/Stroop denotes whether target word was a Non-Stroop or Stroop word). A/B, a three-way interaction among word/color congruency and target Stroop, showing double incongruency slowed down responses only when target was not a Stroop word. C–E, reaction time of word congruent and incongruent conditions in all trials (C), 1st quarter trials (D), and the 4th quarter trials (E) in Experiment 2. Black dots and lines denote longer RT in the incongruent condition while gray dots and lines denote longer RT in the congruent condition. Each pair of dots represents one participant. The bars denote group mean with the error bars indicating standard error of the mean (SEM). Asterisk denotes significance (A: Exp 1: $p = 0.02$ (post hoc 2-tailed paired t test); E: Exp 2 4th quarter trials: $p = 0.04$ (a main effect in a three-way ANOVA)).

(word-color congruent)). In this article, we refer to prime-target color congruency as color congruency, prime-target word congruency as word congruency, and target word-color congruency as target Stroop/Non-Stroop throughout the manuscript. A three-way repeated measures analysis of variance was performed on the target word reaction time (please note that in all experiments the average accuracy on the word or color-naming task was near ceiling (all > 97%). Hence the analyses were focused on the reaction time). In Experiment 1, the main effect of target (reverse) Stroop was found, $F(1, 19) = 37.00$, $p = 0.0000$, $\eta_p^2 = 0.66$, with a main effect of color ($F(1, 19) = 7.63$, $p = 0.01$, $\eta_p^2 = 0.29$) and but not word congruency ($F(1, 19) = 0.73$, $p = 0.40$, $\eta_p^2 = 0.04$). Furthermore, there was an three-way interaction between target Stroop, word and color congruency, $F(1, 19) = 5.81$, $p = 0.03$, $\eta_p^2 = 0.23$. Post hoc comparisons showed that double word-color incongruency significantly slowed down target response only when the target was not a Stroop word (paired $t(19) = -2.43$, $p = 0.02$, marginally

significant after correction), but not when it was a Stroop word (paired $t(19) = -1.96$, $p = 0.06$) (Fig. 2a, b).

The same analysis was performed on the data of Experiment 2 (color-naming). The main effect of target Stroop was also found, $F(1, 19) = 56.01$, $p = 0.0000$, $\eta_p^2 = 0.75$. However, no further main effects or interactions from the prime-target congruency were found (word: $F(1, 19) = 1.36$, $p = 0.26$, $\eta_p^2 = 0.07$, Fig. 2c; color: $F(1, 19) = 2.84$, $p = 0.11$, $\eta_p^2 = 0.13$). Experiments 1 and 2 thus showed that the same set of unconscious stimuli could exert an interfering effect in one context but not another, even with identical time sequence of stimulus delivery and the subsequent influenced target.

**Unconscious word interference modulated by task load.** The sharp differences between Experiments 1 and 2 showed a lack of unconscious interference when the nature of the task was of high load (i.e. the classical Stroop effect), indicating the possibility that

unconscious and conscious processes were competing with attentional resources. However, the color-naming task was also non-semantic in nature, suggesting that a task-driven attentional shift in the conscious task (i.e. Stroop task) could modulate unconscious processing (i.e. prime interference). Therefore, we had two competing attention-related modulation accounts to explain the lack of interference in Experiment 2. We hypothesized that the higher difficulty of color-naming task masked all unconscious information. This is consistent with the load theory which posits attention as general resources utilized and competed between distinct processes[3]. If task load could explain the differences in Experiments 1 and 2, one will predict that under the same difficult color-naming task, when the task difficulty was lowered (e.g. due to practice effect), similar color and word interferences will emerge. To examine this possibility, we further split the trials into the 1st and 4th quarters. A direct comparison between trials in 1st and 4th quarters showed a significant decrease of reaction time (1st: 1437 ms 4th: 1308 ms, t (19) = 3.25, p = 0.00), suggesting a practice effect and a decrease of task load. Identical analysis performed on the 1st quarter trials showed very similar pattern to the overall data: The main effect of target Stroop was found, $F(1, 19) = 21.46$, $p = 0.0002$, $\eta_p^2 = 0.53$, with no further main effects or interaction from the prime-target relationship (word: $F(1, 19) = 0.13$, $p = 0.72$, $\eta_p^2 = 0.01$, Fig. 2d; color: $F(1, 19) = 1.56$, $p = 0.23$, $\eta_p^2 = 0.08$). However, identical analysis performed on the 4th quarter trials showed a main effect of word incongruency $F(1, 19) = 4.62$, $p = 0.04$, $\eta_p^2 = 0.20$ (Fig. 2e), but not color incongruency effect $F(1, 19) = 0.39$, $p = 0.54$, $\eta_p^2 = 0.02$, in addition to the effect of target Stroop, $F(1, 19) = 27.98$, $p = 0.0000$, $\eta_p^2 = 0.60$.

**Replication of unconscious word-induced interference.** Our first two experiments clearly showed that task-induced attentional load modulated the extent to which unconscious stimuli exerted an interfering effect. Experiment 1 showed word-induced and color-induced interference between a suppressed prime and a visible target slowed down the response time to the non-Stroop target. Critically, such interference disappeared when the task was of high load (i.e. color-naming) in Experiment 2. Surprisingly, a further analysis separating early and late trials revealed word-induced but not color-induced interference in the later trials, suggesting that word-induced semantic interference may be more resilient to the current conscious task demands. These findings showed an asymmetry of attentional modulation on prime inter-ferences. To further replicate these findings with a cleaner design, we isolated the word and color components in the later experiments.

In Experiments 3 and 4 we focused only on the word aspect and aimed to re-examine and replicate the word-induced semantic incongruency effect. The experiments had identical trial sequence and design as Experiments 1 and 2, except for two changes. (1) The prime words were made colorless. (2) Half of the primes were made blank. These blank trials later served as our baseline to compare against and allowed us to calculate the percentage reaction time changes between prime absent and present trials.

In Experiment 3 (word naming), the reverse Stroop effect in the target responses was evident (Stroop and non-Stroop trials: t(19) = 4.21, $p = 0.00$, Cohen's $d_{av} = 0.35$). In Experiment 4, Stroop effect was found (Stroop and non-Stroop trials: t(19) = 6.25, $p = 0.00$, Cohen's $d_{av} = 0.58$).

To isolate the effects of congruency/incongruency, prior to comparing between the congruent and incongruent conditions, we first calculated target response time change with against without the prime. Therefore, the following results all appear in reaction time percentage changes from the baseline blank trials. More general ANOVA results in the style of Experiments 1 and 2 are included in supplementary information (Supplementary Note 2). In Experiment 3, a paired t test directly compared

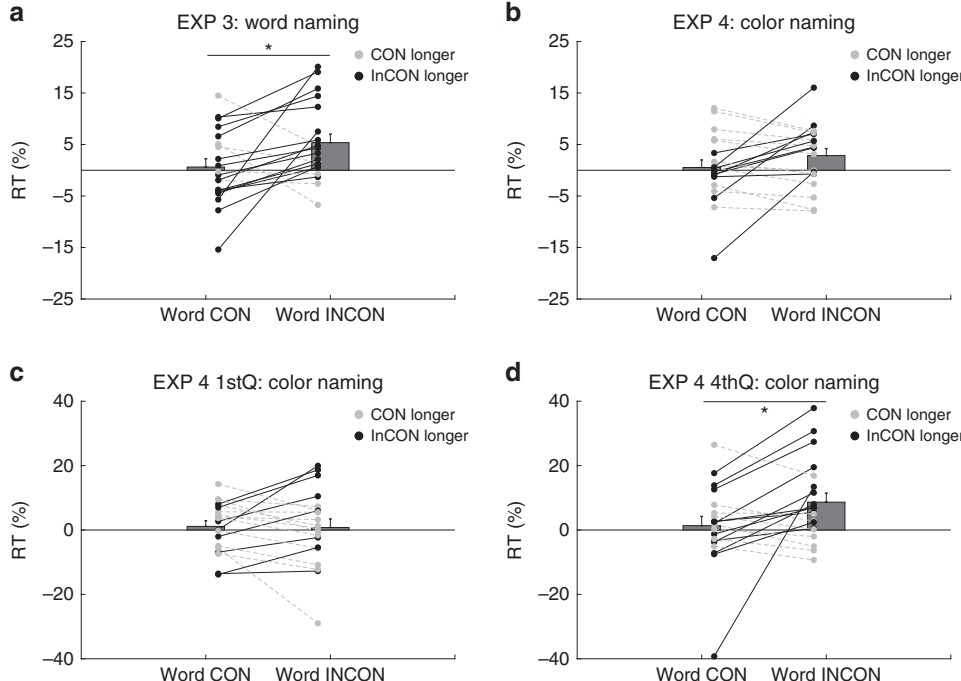

**Fig. 3 Experiments 3 and 4 results.** $n = 20$ in each experiment. *Y*-axis denotes reaction time, and *X*-axis denotes two critical conditions: word congruent (Word CON) and word incongruent (Word INCON). Black dots and lines denote longer RT in the word incongruent condition while gray dots and lines denote longer RT in the word congruent condition. Each pair of dots represents one participant. The bars denote group mean with the error bars indicating standard error of the mean (SEM). Asterisk denotes significance (A: Exp 3: $p = 0.03$; D: Exp 4 4th quarter trials: $p = 0.02$. Both 2-tailed paired t tests).

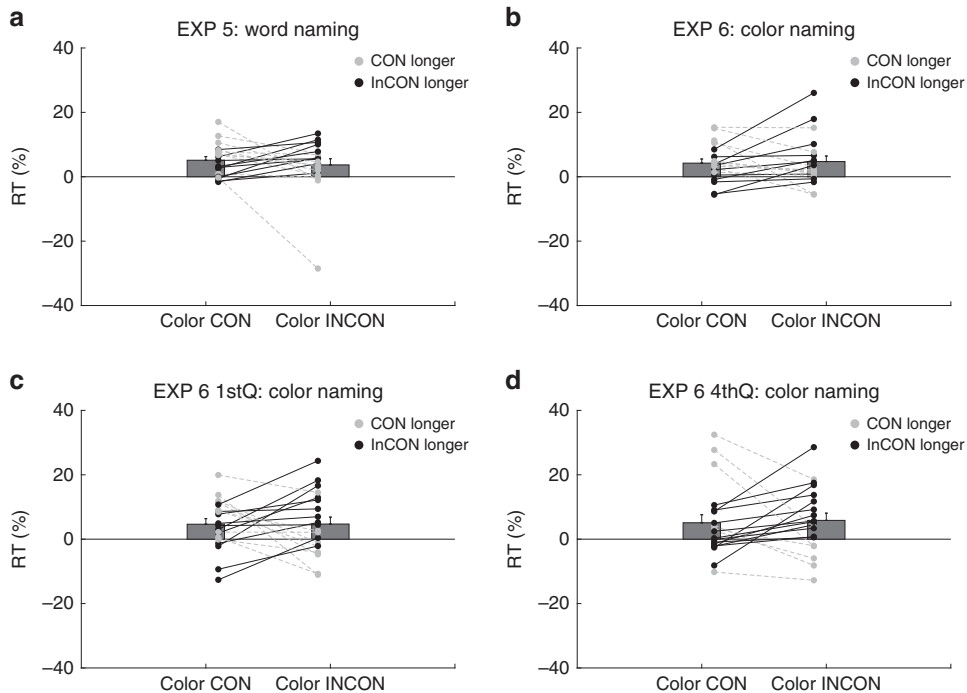

**Fig. 4 Experiments 5 and 6 results.** $n = 20$ in each experiment. Y-axis denotes reaction time, and X-axis denotes two critical conditions: color congruent (Color CON) and color incongruent (Color INCON). Each pair of dots represents one participant. Black dots and lines denote longer RT in the color incongruent condition while gray dots and lines denote longer RT in the color congruent condition. The bars denote group mean with the error bars indicating standard error of the mean (SEM).

between semantically incongruent vs. congruent trials after normalizing against the blank trials. A significant word interference effect was found ($t(19) = 2.43$, $p = 0.03$, Cohen's $d_{av} = 0.65$, Fig. 3a), showing that when the invisible prime and visible target were incongruent, there was a slowing effect on target response of 4.74% (compared to blank trials: word congruent trials: 0.62% slower, word incongruent trials: 5.36% slower).

In Experiment 4 (color naming), a paired t test directly compared semantically incongruent and congruent trials after normalizing against the blank trials showed null results in overall trials ($t(19) = 1.54$, $p = 0.14$, Fig. 3b). Similar to our analysis in Experiment 2, we split the data into 1st quarter and 4th quarter trials. A significant word interference effect was found only in the 4th quarter trials ($t(19) = 2.30$, $p = 0.03$, Cohen's $d_{av} = 0.57$, Fig. 3d) but not in the 1st quarter trials ($t(19) = 0.16$, $p = 0.87$, Cohen's $d_{av} = 0.03$, Fig. 3c). This result again showed that when the task load decreased in a high-load task (color-naming), the incongruency between an invisible prime and a visible target slowed down target response of 7.29% (compared to blank trials: semantically congruent trials: 1.35% slower, semantically incongruent trials: 8.64% slower).

The results of Experiments 3 and 4 replicated what we found in Experiments 1 and 2, showing that semantic interference from an unconscious incongruent word exerted a slowing effect on the subsequent target response. Importantly, this effect was modulated by the conscious task demands. Similarly, the late emerging semantic interference effect was again found under a more demanding color-naming task, when extended practice had reduced the task load.

**No conclusive unconscious color interference**. In the next two experiments, we re-examined the color interference between the invisible prime and the target, which was evident in Experiment 1

with a three-way interaction with word interference and target Stroop but disappeared in Experiment 2. Experiments 5 and 6 set out to examine whether color congruency alone exhibits an interference effect on target response. In these experiments, the primes were colored symbols (i.e. XXXX in blue or red) to cleanly isolate and test the effect of color in the current paradigm.

The reverse Stroop effect and Stroop effects were also evident in Experiments 5 (Stroop and non-Stroop trials: $t(19) = 4.97$, $p = 0.00$, Cohen's $d_{av} = 0.32$) and 6 (Stroop and non-Stroop trials: $t(19) = 6.80$, $p = 0.00$, Cohen's $d_{av} = 0.78$).

In Experiment 5, the RTs in each condition were first normalized against blank trials. No slowing effect was found in the color incongruent trials with $t(19) = 0.7$, $p = 0.49$, Cohen's $d_{av} = 0.21$, Fig. 4a. In Experiment 6, similar to Experiment 2, no color incongruency effect was found in all trials ($t(19) = 0.28$, $p = 0.78$, Cohen's $d_{av} = 0.17$ Fig. 4b) with both conditions showing a slowing effect on target response (compared to blank trials: color congruent trials: 4.23 % slower, color incongruent trials: 4.72 % slower). Such null effect was found in the 1st and 4th quarter trials: (1st: $t(19) = 0.02$, $p = 0.98$,, Cohen's $d_{av} = 0.01$, Fig. 4c, 4th: $t(19) = 0.26$, $p = 0.80$, Cohen's $d_{av} = 0.07$, Fig. 4d) with both conditions showing a slowing effect on target response (compared to blank trials: incongruent vs. congruent trials 1st Q: 4.70 vs. 4.64%; 4th Q: 5.84 vs. 5.10% slower).

These results indicate that non-semantic perceptual stimulus congruency (i.e. color) under interocular suppression did not conclusive yield an interference effect on target response in the current paradigm.

**Attentional modulation: Task load on semantic congruency**. In Experiments 1–4, we found that when the subliminal prime and supraliminal target were semantically incongruent (e.g. BLUE vs. RED), the incongruency prompted slower target responses. This effect was further shown to be modulated by task load. When the

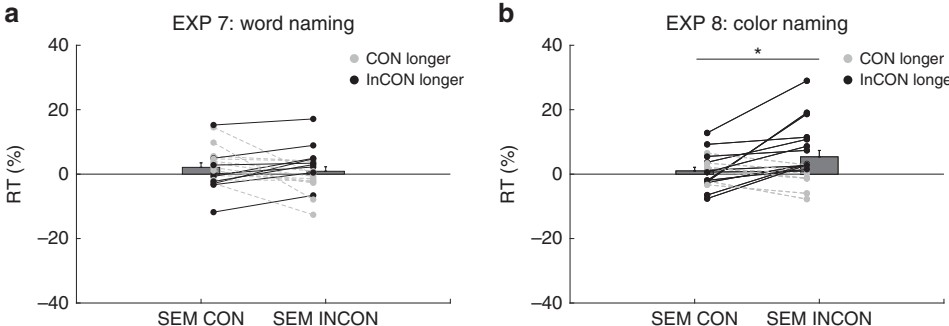

**Fig. 5 Experiments 7 and 8 results.** $n = 20$ in each experiment. Y-axis denotes reaction time, and X-axis denotes two critical conditions: semantically congruent (SEM-CON) and semantically incongruent (SEM-INCON). Black dots and lines denote longer RT in the semantically incongruent condition while gray dots and lines denote longer RT in the semantically congruent condition. Each pair represents one participant. The bars denote group mean with the error bars indicating standard error of the mean (SEM). Asterisk denotes significance (Exp 8, $p = 0.02$, 2-tailed paired t test from prime-target co-localized trials).

task load was low (Experiments 1 and 3, word naming), the effect appeared in all trials. When the task load was high (Experiments 2 and 4, color naming), the effect appeared only in the later practiced trials. This effect was interpreted as a word-induced subliminal semantic interference. However, as the subliminal primes and targets had identical visual forms (BLUE/RED), one may argue that the effect could simply emerge from low-level visual feature adaptation. This was not found in our additional analysis in which we examined whether co-localization of prime-target was critical to the effect (Supplementary Note 4). However, since our unconscious primes and conscious targets shared not only semantic but also orthographic similarities, it is almost impossible to tease apart if such effect was due to word form (orthographic) incongruency or semantic/conceptual incongruency, or both. To directly examine whether task-induced attention could modulate pure semantic interference, we adopted the design in Experiments 3 and 4 with one adjustment: replacing the target words to scarlet and navy. By utilizing these two target words, the prime and target now contained no orthographic similarities while still exhibited semantic/conceptual (in)congruencies.

Furthermore, it has been shown that once the color association between colors and words is weaker (e.g. naming the color blue on the word red vs. and), the Stroop interference from color-naming weakens[21]. The authors also proposed that word frequency could play a key role as low word frequency is typically linked to weaker association with colors. This early observation led to the prediction in our current study that color-naming on the words navy and scarlet would accompany weaker word semantic interference (i.e. weaker Stroop effect) as demonstrated in the original study. Indeed, in our pilot data ($n = 2$), this choice of lower-frequency words as targets reversed the relative task load for word naming and color naming: word-naming was still accompanied by an interference from the color of the target (reverse Stroop effect), while the typical Stroop effect from color-naming disappeared. This opposite pattern of the strength of Stroop/reverse Stroop effects was later confirmed with twenty participants in each experiment. Such pattern teased apart the tight relationship between task load and task set alignment in our previous experiments. Previously, the stronger unconscious prime semantic incongruency effect in word-naming experiments could be attributed to either lower task load or task set alignment (attending to semantic feature). While weaker semantic incongruency effect in color-naming could be due to high task load or task set misalignment (attending to non-semantic feature). Thus, it was again almost

impossible to pinpoint whether task-induced attentional load or feature-specific attention modulated the unconscious effect. Nonetheless, with the current design, we have created a high-load × task-effect aligned feature-attending (word naming) and low-load × task-effect misaligned feature-attending (color-naming). The new experiments allowed us to examine two separate pairs of conflicting hypotheses on (1) the modulated feature: orthography or semantics and (2) the modulating source: task load or task set. In (1), if pure semantics from an unconscious prime could exert an effect, we expected to see unconscious interference on the target responses. Otherwise if the interference effect was mainly driven by orthographic similarities, the effect would disappear. In (2), if task load played a key role in modulating the interference, the interference would appear in the new low-load experiment (Experiment 8, color-naming) but not the new high load experiment (Experiment 7, word-naming, or only in the later practiced trials). Otherwise if task set alignment played a key role, the interference would appear in the word-naming but not the color-naming experiment.

The critical difference was that the reverse Stroop effect was evident in Experiment 7 (Stroop and non-Stroop trials: t(19) = 3.97, $p = 0.00$, Cohen's $d_{av} = 0.32$) while the Stroop effect was not significant in Experiment 8 (Stroop and non-Stroop trials: t(19) = 2.04, $p = 0.06$, Cohen's $d_{av} = 0.16$).

In Experiment 7, the RTs in each condition were first normalized against blank trials. A direct paired comparison between semantically incongruent and congruent trials showed null effect (t(19) = −0.82, $p = 0.42$, Cohen's $d_{av} = 0.19$, Fig. 5a). On average, compared to the blank trials, semantic congruent trials were 2.1% slower while incongruent ones 0.9% were slower. We also examined whether prime-target location (same-different) had interacted with the semantic congruency effect. A two-way (semantic congruency; prime-target location consistency) repeated measures analysis of variance was performed on the normalized target word reaction time. Neither the main effect of semantic congruency, $F(1, 19) = 0.09$, $p = 0.77$, $\eta_p^2 = 0.005$, nor the main effect of location congruency was found, $F(1, 19) = 1.32$, $p = 0.26$, $\eta_p^2 = 0.07$. There was no interaction between the two, $F(1, 19) = 0.21$, $p = 0.65$, $\eta_p^2 = 0.01$. Similar results were found in the 4th quarter trials (direct comparison: t(19) = 1.21, $p = 0.24$, Cohen's $d_{av} = 0.33$; semantic congruency × location consistency 2-way ANOVA: main effect of semantic congruency $F(1, 19) = 0.10$, $p = 0.75$, $\eta_p^2 = 0.005$; main effect of location: $F(1, 19) = 0.14$, $p = 0.72$, $\eta_p^2 = 0.007$; interaction: $F(1, 19) = 1.60$, $p = 0.22$, $\eta_p^2 = 0.08$).

**Table 2 Conditions, manipulations, and effects in all experiments.**

| Experiment | Prime | Target | Task | Load | Interference |
|---|---|---|---|---|---|
| 1 | BLUE RED (colored) | BLUE RED (colored) | Word-naming | Low | Yes |
| 2 | BLUE RED (colored) | | Color-naming | High | Yes after practice |
| 3 | BLUE RED | | Word-naming | Low | Yes |
| 4 | BLUE RED | | Color-naming | High | Yes after practice |
| 5 | XXXX (colored) | | Word-naming | Low | No |
| 6 | XXXX (colored) | | Color-naming | High | No |
| 7 | BLUE RED | navy scarlet (colored) | Word-naming | High | No |
| 8 | BLUE RED | | Color-naming | Low | Yes |

The table summarizes all eight experiments where we manipulated the color, word, and semantic relationship between the subliminal prime and supraliminal target. Task (word-naming vs. color-naming) and task load (high vs. low, according to the strength of Stroop/Reverse Stroop effect) in each experiment are presented.

In Experiment 8, a direct paired comparison on the normalized RT between semantically congruent and incongruent trials showed null effect (t(19) = −1.12, $p = 0.28$, Cohen's $d_{av} = 0.24$). On average, compared to the blank trials, semantic congruent trials were 1.32% slower while incongruent ones 2.7% were slower. However, a two-way (semantic congruency; prime-target location consistency) repeated measures analysis of variance showed an interaction between the prime-target location and semantic congruency with $F(1, 19) = 8.13$, $p = 0.01$, $\eta_p^2 = 0.30$. The main effects were not significant (semantic congruency: $F(1, 19) = 1.1$, $p = 0.29$, $\eta_p^2 = 0.06$; location congruency: $F(1, 19) = 0.05$, $p = 0.83$, $\eta_p^2 = 0.003$). A planned post hoc comparison between semantic congruent and incongruent trials when the prime-target were co-localized showed a significant effect with t (19) = −2.45, $p = 0.02$, Cohen's $d_{av} = 0.63$, Fig. 5b). When the prime and target were co-localized, semantic congruent trials were 0.98% slower while incongruent ones were 5.36% slower.

These results showed that, without orthographic similarities, the semantic incongruency between the prime and target did pose a slowing effect on the target responses if they were co-localized. The observation that the semantic effect further interacted with prime-target co-localization indicated the need of spatial attention toward the primed location for such effect to occur. This semantic effect was also modulated by the task load. In Experiment 8 (color-naming), when the task load was low as indicated by the lack of the Stroop effect, the semantic incongruency slowed down target response. In contrast, in Experiment 7 (word-naming), when the task load was high as indicated by the reverse Stroop effect, such effect disappeared. As task alignment and task load were separated in these two experiments (i.e. Experiment 7: semantic task but high load, Experiment 8: non-semantic task but low-load), distinct from all previous experiments. This particular finding stressed that the level of task load, but not task alignment, modulated an unconscious semantic effect in our paradigm. Table 2 summarizes the stimulus type, task, and effect in all experiments.

**General discussion.** Can attention operate on high-level unconscious visual information? The findings from our double Stroop paradigm gave an affirmative answer, showing that word-induced semantic incongruence between an interocularly suppressed subliminal prime and a subsequent supraliminal Stroop target slowed down target responses. Critically, this semantic interference was modulated by task load. The word-induced semantic incongruence exhibited a slowing effect on target response when the task was of low load (word-naming, Experiments 1 and 3). This effect disappeared when the task was of high load (color-naming, Experiments 2 and 4). Further analysis strengthened this

account by showing that in the later trials of the high load experiments, when load became decreased due to a significant practice effect, the semantic incongruency effect re-emerged (4th quarter trials, Experiments 2 and 4). To sum up, our findings show a strong attentional gating on unconscious semantic information to different levels. Word semantics is more automatically activated, as posited in the classical Stroop phenomenon[20], and is thus more resilient to attention deprivation. Accordingly, subliminal semantic incongruence leads to task interference even under high load tasks. On the other hand, we only observed a color incongruency effect as part of the combinatory word-color double incongruency effect in Experiment 1, and subsequent experiments did not yield significant color incongruency effect, suggesting that unconscious color information was inconsequential in eliciting a strong interfering effect in the current paradigm.

Two important questions were left unanswered after our first four experiments. Firstly, although the general predictions of the load theory are compatible with our findings, a feature-based explanation is also possible. Task-induced attention might selectively trigger different cognitive sets under different tasks, which in turn would gate how an unconscious prime interfered with subsequent target performance. For example, in the word-naming experiments (Experiments 1 and 3), participants' attention was selectively oriented to the word semantics. Such attention tuning would have been applied further to the suppressed prime, allowing semantic interferences to occur. In contrast, in the color-naming experiments, attention was deployed to the word color, which would have interfered the semantic processing of the target (i.e. the Stroop effect) as well as hindered the semantic interferences from the suppressed prime. Since our interfering effects were semantic in nature, word-induced semantic incongruency effects could have been gated by this cognitive-set-induced-tuning to different levels: an incongruent word interference was evident in word-naming experiments; while under color-naming task, such effect only re-emerged after the participants became more fluent on the task (i.e. 4th quarter trials in Experiments 2 and 4). We also provide a discussion regarding the classical negative priming in the Stroop paradigm in Supplementary Note 3. Secondly, one could argue that since identical word-forms were used as primes and targets, what appeared to be a semantic incongruency effect could be driven merely by orthographic dissimilarity. The initial experiments did partially address this by jittering the prime-target locations and font sizes, and our further analyses focusing on the interaction between the semantic effect and location yielded null effect, which excluded the possibility of low-level adaptation (Supplementary Note 4). Nevertheless, this does not distinguish between orthography and semantics.

Thus, (1) to ensure that a pure semantic effect could be modulated by attention, and (2) to tease apart cognitive-set-induced-tuning and task load modulation, we went one step further in Experiments 7 and 8. We changed the targets to "navy" and "scarlet", removing the orthographic similarity of the prime and target while maintaining the semantic relationship. Importantly, this caused a reversal in relative task load: For the color-naming task, the Stroop effect disappeared–showing that it was now a low-load task; for word-naming, the reverse Stroop effect remained–it was now relatively high-load. This allowed us to make a double-dissociation between cognitive-set-induced-tuning attention and task load, enabling us to pinpoint the determining factor in modulating the unconscious effect. The results showed the unconscious semantic incongruency effect when the task load was low yet non-semantic (Experiment 8, color-naming) but not when the task load was high but semantic (Experiment 7, word-naming). This indicates that general availability of attentional resources, but not cognitive-set-induced-tuning attention, played a key role in modulating the unconscious semantic interference in our study.

Our findings that the word semantics of the subliminal word elicited interfering effects while being modulated by task-induced demands provide several novel insights to the field of unconscious processing. Firstly, as color-naming is considered a more difficult task and less automatic in a typical Stroop task[20], one can posit that (1) when the task load is high, the amount of attention required to elicit an unconscious effect is constrained (Experiment 2 and 4). The load theory[2,3] makes clear predictions on how attention modulates both conscious and unconscious processes, particularly regarding attention as limited resources facing constant competition among different perceptual processes. In fact, the load theory clearly points out that "…the effects of attention on unseen (or seen) ignored stimuli will only be observed when resources are sufficiently engaged by another task or stimulus and thus unavailable to the stimulus in question."[3]. To further this claim, we believed that a strong paradigm not only should show the disappearance of an unconscious effect with high task load but also should bring the same effect back while the task load decreases as a significant practice effect emerges. Our double Stroop paradigm proved to serve this purpose, showing a clear unconscious word-induced semantic interference in the low-demanding condition (word-naming). In the high load condition (color-naming), the effect disappeared as expected. What is critical, as predicted by the load theory, is that when the task demand decreased due to extended practice, the unconscious effect gradually regained its existence. However, we have to point out that the location of attention gating in the perceptual/neural pathway remains elusive. For instance, high task load could impose a higher need for attentional resources for the conscious stimulus and thus leave lesser attentional resources for the unconscious stimulus in a general manner. That is, the effect of task load was effective on the whole experimental block rather than at the single trial level. It is plausible that the gating occurred during the presentation of the unconscious stimulus, even prior to the target presentation, which was where the load was imposed. On the other hand, the gating could have occurred at a much later stage during the interference between the unconscious and conscious stimuli. That is, the semantic effect was modulated only when the load inducer (i.e. the conscious target) was present. Future experiments are required to examine these possibilities.

As unconscious processing is classically associated with automaticity, attention is thought to play little role in such automatic unconscious effects. However, this view has been recently challenged by a series of experiments targeting semantic priming under the masking paradigm. Kiefer and Martens[22] showed that a masked word elicited stronger semantic priming when it was preceded by a semantic task (e.g. judging whether a word denotes living or non-living object), compared to a perceptual task (e.g. judging the shape of the letter). The results showed less behavioral semantic priming (but still significant), and eliminated N400 effect in the perceptual task condition. The attenuated yet still significant behavioral semantic priming effect under a perceptual task was also found in another study[23]. In our study, the relative strength of Stroop and Reverse Stroop effects (with regard to semantics and color interference within the supraliminal target word) indeed confirmed the high/low load manipulation. This paradigm allowed simultaneous examination on the attentional modulation on word- and color-induced semantic interferences, gauging how attention gates concurrent processes. For instance, the residual subliminal semantic priming even when task-induced attention was deployed to another visual feature (i.e. color) was also seen in our data (Experiments 2 and 4, 4th quarter trials), suggesting that unconscious word-induced semantic processing, though could be modulated by conscious task demand, resumed its effect when the current task demand decreased. Moreover, in Experiments 7 and 8 where we removed the orthographic similarities of the prime and target while maintaining their semantic relationship (BLUE-navy; RED-scarlet), prime-target semantic incongruency still posed a slowing effect when the task load was low (color-naming) and the prime and target were co-localized. This effect disappeared when the task load was high (word-naming) and even in the later trials, suggesting that pure unconscious semantic effect was further in need of attentional resources, both in the general domain (regarding task load) and spatial domain (regarding prime-target location adjacency).

The relationship between attention and consciousness has been a heated debate involving researchers in Neuroscience, Psychology, and Philosophy. Koch and Tsuchiya[24] posit that attention and consciousness are doubly dissociable, that is, one can show conditions where attention is summoned but consciousness is yet to emerge and vice versa. In contrast, Cohen et al. propose that attention gates consciousness, arguing that a stimulus/scene/object only enters our consciousness when some amount of attention is deployed to the item[25]. Thus, according to this attention gating theory, there is a causal relationship between attention and consciousness. Our data exhibit attention without consciousness, showing task-induced attentional modulation on unconscious processes. Together with previous evidence showing that bottom-up attention can be directed to an unconscious stimulus (e.g. salient singleton[18]; random motion[26]; attractive face[27]), our results further show that task-induced top-down attention constraints an interfering effect elicited by an unconscious stimulus. However, this interpretation is not necessarily at odds with the attention gating consciousness theory as these results can be seen as an expansion of the attention gating system from the realm of consciousness to unconsciousness, which is previously acknowledged[25].

Since the invention of continuous flash suppression[28], whether high-level subliminal information survives strong interocular suppression has been a matter of debate. Jiang et al.[29] reported familiarity effect from both face and linguistic stimuli, showing that upright faces broke through interocular suppression faster than inverted faces. Similar faster breaking time was shown between words from one's own native language and words from a foreign language. More directly, Costello et al.[30] showed that a word broke through suppression faster if the preceding word was semantically or orthographically related, compared to the condition in which the suppressed word was preceded by an unrelated word. Recently, Hung and Hsieh[31] showed that subsequent to a sential context, syntactically incongruent words break through interocular suppression faster than the congruent counterparts.

However, these studies potentially suffer from the disadvantage of the breaking-suppression paradigm[32], unable to distinguish a pure unconscious effect from an access-to-consciousness effect. That is, as breaking-suppression relies on the conscious detection of a suppressed stimulus, it is thus hard to tease apart the actual origin of the effect. Current study provided a cleaner paradigm in which the interference effect was assessed by how a precedent suppressed word (prime) affected the following target response even when the participants had not broken suppression and performed at chance on localizing the suppressed stimulus. Therefore, the semantic interference from an interocularly suppressed word provided clear evidence for high-level subliminal semantic processing. Moreover, as our findings show that the attentional requirement of a task serves as a gating mechanism to an unconscious effect, it is worth reconsidering the inconsistent results from recent interocular suppression studies under this framework. Specifically, subliminal stimuli very often suffer from poor stimulus signal-to-noise ratio, even more so in the interocular paradigm as experimenters actively suppress the visibility of a dim and static stimulus. It is thus difficult to distinguish a true null finding where a subliminal stimulus does not intrinsically elicit an effect from a more trivial explanation: the underpowered nature of subliminal stimuli. Here we provide evidence for another critical factor: the necessity of attention.

As our study showed that attention load of the task modulated the semantic interference from an unconscious prime, it is important to point out that previous studies have shown that attention also interacts with how stimulus invisibility is achieved under a visual suppression/masking paradigm[33,34]. For example, it has been shown that semantic information is registered when the invisibility is induced by the lack of attention under interocular suppression[34]. How exactly did the task-induced attention load, and any other factors related to task, interacted with interocular suppression in our study will require future studies. However, to ensure that stimulus invisibility was properly achieved in our experiments, we have proceeded to perform additional analyses on our objective awareness test (i.e. 2AFC prime location task). Apart from having near-ceiling performance on visible and blank catch trials as well as group-level chance performance on the 2AFC location task, we ran a binomial test on the responses of the 2AFC location task on every single participant and analyzed our data only on those participants that showed chance performance[35]. The results remained largely identical (see Supplementary Note 1). Whereas measuring stimulus visibility always has its unavoidable uncertainty, we believe our results showed converging, realistically achievable evidence of the prime invisibility.

The current findings shed light on the attentional gating of subliminal information, expanding the capacity limitations of attention outside the realm of consciousness. More critically, we show that such attentional modulation occurs with high-level semantic information, opening up the possibility that the attentional load of concurrent task modulates a wide spectrum of information processing in the absence of consciousness. If attention is a limited resource that is shared and competed for, not only by what we are conscious of but also what we are not, the unconscious processes are potentially constrained by our conscious deliberations. Real-world implicit influences in the surroundings are thus gated by how we distribute our attention, indicating a mechanism in which unconscious contents can be consciously and voluntarily enhanced or weakened.

## Methods

**General experimental apparatus**. In all experiments, the visual stimuli were generated with MATLAB (The MathWorks, Inc., Natick, MA) and PsychToolbox[36,37]. Participants viewed the dichoptic images through a mirror stereoscope and rested on a chin rest, from a distance of 42 cm. The stimuli were presented against a black background on a 30-in. Apple M9179LL/A LCD monitor with a resolution of 2560 × 1600 pixels and a refresh rate of 60 Hz. Throughout the experiment, a white frame (subtending 5.4° × 5.4°) remained on-screen to facilitate proper fusion.

**Participants**. Throughout all experiments, all participants (age range: 18–36) reported normal or corrected-to-normal vision. They reported no history of language deficits and were proficient in English. They gave written informed consent prior to the experiment and were reimbursed $15 for participating in a 60-min session. This study was approved by the institutional review board of the California Institute of Technology. Participants that (1) had performance 3 standard deviations away from the group mean on the catch tasks or/and (2) failed to achieve successful calibration on prime luminance or/and (3) brokethrough suppression yet failed to indicate prime location (see below experimental design and procedure) were removed before entering analysis. (see Stimuli and Procedure of each experiment; Experiment 1: $n = 3$; Experiment 2: $n = 2$; Experiment 3: $n = 1$; Experiment 4: $n = 2$; Experiment 5: $n = 0$; Experiment 6: $n = 0$; Experiment 7: $n = 1$; Experiment 8: 2). A total number of 20 participants was targeted in each experiment based on our 80% power calculation in our previous study[31]. All participants were naïve to the purpose of the experiments.

**Reaction time data pre-analysis processing**. All trials that had reaction time longer than 5 s or shorter than 500 ms were pre-excluded. Furthermore, the reaction time data underwent per-participant per-condition outlier removal to remove data points 3 standard deviations away from the average.

**Experiments 1 and 2: experimental design and procedure**. Prior to the experiment, participants' eye dominance was determined by the Miles test[38]. Two words and two colors were selected to create word-color (in)congruency in the prime and target: word BLUE or RED in color blue or red. Each trial began with a blank screen lasting for a varied SOA ranging from 0.1 to 1 s. After which the dominant eye received a series of colorful flashing Mondrian suppressors consisted of orange, yellow, green, indigo, and violet. No blue and red colors were in the suppressors to prevent confusion between the suppressor and the suppressed. On the non-dominant eye, the prime was presented with the contrast ramping up from 0% to the designated contrast determined by a trial-by-trial thresholding procedure. Each color had its own 3-up-1-down contrast calibration staircase: when a suppressed stimulus was detected, the contrast decreased in the next trial, while if a suppressed stimulus was not detected three trials in a row, the contrast increased in the next trial. The initial contrast was chosen according to participants' performance on the 20 practice trials before they proceeded the actual experiment.

Both the suppressor and the suppressed were presented in an on-and-off manner with 400 ms on and 400 ms off to ensure stronger suppression. During 400-ms stimulus presence, the suppressed was sandwiched by the suppressor temporally, leaving the first and last two frames absence to prevent sudden breakthrough and afterimage, respectively. This led to a 333-ms prime presentation time in each on-and-off cycle. This on-and-off cycle lasted five times, resulting in 4-s suppression period (in a recent study (Hung and Hsieh, under review), we showed that intermittent presentation, coined discontinuous flash suppression, during the suppression period delayed breakthrough of the stimulus and hence potentially increased subliminal signal of the stimulus. Thus we applied such interocular suppression paradigm here to achieve longer suppression and exposure duration of the prime). The suppressed prime was presented either above or below the fixation point. The location was counterbalanced across the conditions. After which, the target was presented to the suppressor eye in a different font size and a slightly jittered location to prevent simple adaptation until response.

In each trial, the participant had three tasks in sequence. All completed via button press. (1) During the suppression period, a detection task was given. The participant was instructed to respond as soon as any part of the word stimulus had been detected. It was stressed that this is the most important task and even seeing a stroke of the word counted as detection. If the prime had been detected, the trial ended immediately. (2) If the prime was not detected, the participant had to name the word in Experiment 1 and the color in Experiment 2 of the visible target word. It was stressed that both speed and accuracy were important. (3) At the end of the trial, a two-alternative-force-choice (2AFC) location task asked the participant to judge the location of the suppressed prime. It was made clear that they should try their best to judge even if the prime remained undetected in the suppression period. Please note that this 2AFC location task did not require the participant to hold prime information until the end of the trial but instead served as a reconfirmation of prime invisibility. Thus, the prime invisibility was assessed with both an immediate subjective and a retrospective objective criterion.

In total, 352 trials were completed with 320 experimental trials (2 prime colors × 2 prime words × 2 target colors × 2 target words × 2 prime location × 10 repetitions) and 32 catch trials, including 16 blank trials where no stimulus was delivered during the suppression period and 16 visible trials where the stimulus was delivered to the dominant eye and superimposed with the Mondrians. These catch trials allowed us to further gauge if participants developed any responses bias in the course of the experiment.

**Experiments 3 and 4: experimental design and procedure**. The experimental design and procedure of Experiments 3 and 4 were identical to Experiments 1 and 2 except for two alterations: (1) The prime words were made colorless. We focused only on the semantic effect and aimed to re-examine and replicate the semantic incongruency effect of the suppressed prime and visible target. (2) Half of the trials were made blank. In total 336 trials were performed in each individual (2 prime presence/blank × 2 prime words × 2 target colors × 2 target words × 2 prime locations × 10 repetitions + 16 visible catch trials). These blank trials not only allowed us to gauge the false alarm rates of breaking suppression but also allowed us to normalize the reaction time of suppressed prime presence trials against.

$$RTnomalized = \frac{RTwith\ prime - RTwithout\ prime}{(RTwith\ prime + RT\ without\ prime)/2}. \quad (1)$$

Participants in Experiment 3 were asked to name the word of the target while participants in Experiment 4 were asked to name the color of the target.

**Experiments 5 and 6: experimental design and procedure**. The experimental design and procedure of Experiments 5 and 6 were identical to Experiments 1 and 2 except for two alternations: (1) The primes were meaningless symbols with color (XXXX in color blue or red). (2) Identical to Experiments 3 and 4, half of the trials were made blank. In total 336 trials were performed in each individual 2 prime presence/blank × 2 prime colors × 2 target colors × 2 target words × 2 prime locations × 10 repetitions + 16 visible catch trials.

**Experiments 7 and 8: experimental design and procedure**. The experimental design and procedure of Experiments 7 and 8 were identical to Experiments 3 and 4 except for one alternation: the targets were replaced by scarlet and navy so that the prime-target relationship was semantic but not orthographic. Similarly, in total 336 trials were performed in each individual 2 prime presence/blank × 2 prime colors × 2 target colors × 2 target words × 2 prime locations × 10 repetitions + 16 visible catch trials.

**Reporting summary**. Further information on research design is available in the Nature Research Reporting Summary linked to this article.

## Data availability
The raw data are available from the corresponding author upon reasonable request.

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

## Acknowledgements
The authors thank the support of James Boswell Postdoctoral Fellowship and Caltech Biology and Biological Engineering Divisional Postdoctoral Fellowship to S.-M.H and the research funding from the Japan Science and Technology Agency (JST) (JST. CREST2014) to S.S.

## Author contributions
Conceptualization, S.-M.H. D.-A.W. and S.S.; Methodology, S.-M.H.; Software, S.-M.H.; Investigation, S.-M.H.; Writing—Original Draft, S.-M.H.; Writing—Review and Editing, S.-M.H. D.-A.W. and S.S.; Supervision, S.S.

## Competing interests
The authors declare no competing interests.

**Additional information**

