## [Peer Review File · Nature Communications]

Reviewers' Comments:

Reviewer #1:

Remarks to the Author:

Summary

Hung et al. evaluated the impact of unconscious, high-level processing with a novel task in which colored words were presented in a sequence where the prime stimulus was rendered invisible owing to interocular suppression. Participants either performed a word-reading task or a color-naming task for the visible word. Hung et al. found that the prime modulated the task performance in a task-specific manner. Specifically, suppressed word modulated the processing of the visible target word in word reading task but the impact of suppressed color was limited in the color-naming task.

Evaluation

It seems that the authors investigated an important question, to what extent the unconscious information is registered and how that is modulated by the task in relation to attention. Despite the small differences in key measures, the authors replicated the results over different stimulus conditions. I believe the claim would contribute to the literature in studies of attention and consciousness. However, it was really difficult to understand the manuscript. As a result, I was unable to evaluate the manuscript properly and I do not rule out a possibility that my concerns below could have been raised based on an incorrect understanding of the study.

1. Clarity of the manuscript

I found that the nature of the experiment was complex, where congruency of the prime and the target was independently manipulated, and the task, the word-reading, and color-naming, was varied across different experiments, and only a subset of conditions was pulled together for comparison. Despite its complexity, the manuscript was not clearly written and conditions were not clearly described, especially Exp 1 and 2. While the experimental conditions were straightforward to the authors, they are not for the readers.

In addition, some terms make the manuscript even more difficult to read. For example, What are Stroop and non-Stroop? It appears that the authors used "Stroop" when the color and word were incongruent, but I was not sure. Congruency and consistency were also mixed. Considering that congruency of the color and word was manipulated for the prime and the target independently, congruency can be applied to the prime, target, or the prime-target relation.

So, please make the manuscript easy to read and check out the PDF whether the figure was properly placed.

2. Invisibility of the prime stimulus

In the literature, first, it is difficult to establish whether the invisible was really invisible because our judgment was susceptible to many factors including criterion and task demand (Yang et al., 2014). Second, distinguishing different types of invisibility matters (Kanai, Walsh & Tseng, 2010). This second issue is particularly important because attention interacts with interocular suppression in making the stimulus suppressed and, thus, its impact to unconscious processing depends on those two types of invisibility (Eo et al., 2016).

Eo, K., Cha, O., Chong, S. C., & Kang, M.-S. (2016). Less Is More: Semantic Information Survives Interocular Suppression When Attention Is Diverted. *The Journal of Neuroscience* :36(20), 5489–5497.
Kanai, R., Walsh, V., & Tseng, C.-H. (2010). Subjective discriminability of invisibility: A framework for distinguishing perceptual and attentional failures of awareness. *Consciousness and Cognition*, 19(4),

1045–1057.

Yang E, Brascamp J, Kang MS, Blake R (2014) On the use of continuous flash suppression for the study of visual processing outside of awareness. *Front Psychol* 5:724.

In Huang et al., the task difficulty changed between word reading and color naming. Participants performed the word-reading/color-naming task as well as the location judgment task, and they had to monitor whether the prime stimulus broke suppression whose location was uncertain. It is, therefore, possible that the prime could have been consciously registered but it was not reported in the experiment because the participants performed several tasks together and, thus, they could have adjusted their criterion more conservatively so that only the obvious, catch trials here, can be reported. Also, note that the previous studies have pointed out that information is registered when the invisibility is induced by the lack of attentional amplification.

3. Theoretically,

Attention can interact with consciousness. I do not have any problem with this claim. However, it is difficult to take the claim for several reasons.

First, Hung et al. did not directly manipulate the load. Instead, they varied the task. If task changes, task set should change accordingly throughout the experiment. The changed task set can influence several aspects of the experiment.

For example, in the word-naming, it is possible that the participants looked into a set of orthographical features of RED and BLUE, instead of processing word RED and BLUE, because there were only two colors and two words. Considering that there was converging evidence that low-level features are registered for the suppressed stimulus, the observed effect could be due to the feature processing rather than high-level semantic processing.

Second, I am not still sure why the suppressed word can slow down the target word processing. The effect was very small as well. One way to demonstrate the presence of the effect was to do a task where both the prime and the target are visible. I believe establishing the double Stroop procedure with the visible stimuli would be important in building an argument, especially for discussing the cognitive processes involved in the task-specific priming effect as well as for providing some baseline level performance we can expect from the procedure.

Third, in a related vein, negative priming has been extensively discussed in the Stroop literature. I think the effect reported in the study seems to have some relevance to this literature. Authors may find some insight or information from the literature.

4. Other concerns and questions.

- How did the authors calculate the accuracy of the blank condition? The task was a 2AFC location judgment task instead of a presence/absence judgment.

- The interocular suppression phase was repeated five times. What was the design goal of this procedure?

- In Experiment 5, one subject data look quite abnormal. Without priming, RTs were much slower. Any thought?

Min-Suk Kang

Reviewer #2:
Remarks to the Author:

This manuscript tackles an interesting question: when attention acts on an unseen stimulus does it only affect low-level properties that may have been extracted in early sensory areas, or does it also affect the processing of higher-level semantic properties associated with the stimulus? There is already evidence that attention can act on stimuli that remain unseen, both in terms of enhancing their processing and in terms of selection (the outline of an unseen object can act as the boundary in object-based attention, Norman et al, 2013, Psych Sci.). The experiments presented here are based on the Stroop effect and examine both forward and reverse colour-word Stroop tasks. The effects of both the colour and word-identity of unseen masked primes on reaction times to consciously perceived Stroop stimuli are reported. We already know that unseen colours can act as primes and that their effect can be modulated by attention (Kentridge et al, 2008, Neuropsychologia) and that unseen words can act as primes (see e.g. Rohaut & Naccache, 2018, EJM) so there are good grounds to expect effects of unseen primes on the Stroop task. The most interesting aspect of the design is that it tests whether the 'ink' colour of an unseen prime influences reaction time in a Stroop task by virtue of its semantics and whether this effect is modulated by attention. Even if it is modulated by attention (and it is), interpretation of the finding is difficult. Should one regard attention as only acting at the stimulus level? For example, is it the processing of the blueness of the ink that is enhanced by attention, but after that, the evocation of blue semantics is only stronger because 'ink blueness' is more strongly represented than it would be should the ink colour have been unattended? Alternatively, should we regard attention itself having enhanced representation of all aspects of the stimulus, both semantic and otherwise?

I found it quite hard to understand the precise procedure of these experiments. The legend to figure 1 makes it clear that the location task was 2AFC and that it occurred after the naming task. It also implies that there was a separate opportunity for subjects to report breakthrough from masking. Reports of breakthrough (rather than correct location responses) resulted in the end of a trial. In the methods section on line 579. I presume this is what is referred to as the 'Detection task' in figure 1 (although this term is never used outside the figure itself). I am not clear what the subjects did on the final cycle of priming in a trial (i.e. when they had not reported breakthrough in the detection tasks). Logically they should have also completed a detection task after the final prime presentation. As colour or word naming in the Stroop paradigm is necessarily a reaction time task, I can't see how preceding this with a prime detection task makes any sense. However, on line 581 the authors report that 'If the prime was not detected, the participants reported the word (Exp 1) or colour (Exp 2) of the visible target. This implied that the detection task was completed prior to word or colour naming.

It is far from clear what the factors 'colour consistency', 'word consistency' and 'target (non)Stroop' mean? I deduced that 'colour consistency', 'word consistency' must mean consistency between the colour/word of the prime and the colour/word of the target and that 'target (non)Stroop' indicates whether the word and colour of the target word differ (Stroop) or match (non-Stroop). There are many other possibilities though. The manuscript would be far more readable if the meaning of these factors was spelled out.

In Experiment 1 there is an interaction between 'target (non)Stroop' and 'colour consistency' but not with 'word consistency'. That suggests that the prime can act to enhance the effect of the Stroop distractor dimension (colour in a naming task), but it doesn't affect the task dimension (word naming in this case). If the colour of the prime and the colour of the target both don't match the target word

in a word naming task RTs are slowest. If the colour of the prime and the target match and the target is non-Stroop – so the colour of the prime and target match the target word, RTs are fastest. I am not sure why this is especially surprising if word reading of the target is so effortless and automatic as to be unaffected by the prime (such reading automaticity is meant to be the basis of the Stroop effect).

In Experiment 2, the classic Stroop colour naming task, this is a Stroop effect, but there are no effects of colour or word consistency between prime and target. Colour naming is effortful and far from automatic. We might expect the name of the target word to be processed automatically (that is what generates the Stroop effect) and this to be so fast that it cannot be made faster by a matching prime or slowed by a distracting one. Once colour naming becomes more practiced the effect of the distractor dimension (word name) becomes stronger (reaching statistical significance). So here when the primary task is effortful, we find no effect of primes, but once it becomes easier the distractor dimension primes exert an effect. Exactly as they do in the word-reading task.

Something has gone wrong around lines 215 to 233. Some blocks of text appear twice. There is a non-sequitur at line 223 (I don't think 'colour naming...' really follows on from its preceding text).

In the discussion of this section the authors claim that "Experiment 1 showed both word-induced and colour-induced semantic interference between a suppressed distractor and a visible target" (line 251). I don't see the evidence for an effect of word consistency on the (reverse) Stroop effect in this experiment. There is no interaction between 'colour-consistency' and 'target (non) Stroop'. There is an interaction between word and colour consistency of the primes, but that effect occurs independent of the Stroop effect. The post-hoc comparisons reported show that it is colour congruency that leads to very slow RTs for Stroop targets and very fast RTs for non-Stroop targets. Word-consistency is playing no role that I can see.

Experiments 3 and 4 repeat the preceding Stroop and reverse-Stroop experiments but use colourless (grey I suppose) primes in order to distinguish between the effects of prime colour and semantics. In addition, half of the trials contain no prime and so can be used as a baseline (although the mere presence of a prime might have an effect of RTs via alerting, so perhaps a better baseline control would use irrelevant words or non-word primes). In the reverse Stroop experiment (Expt 3) there is a Stroop effect and also an effect of prime congruency. I don't know why these analyses are not presented as an ANOVA with factors 'prime congruency' and 'target (non) Stroop', as before. An effect of prime congruence shows that matches between the prime word and target word speed RTs regardless of the colour in which the target word is written. Word-naming is the task dimension of this reverse Stroop task and so we might be surprised that, in contrast to the previous experiments, we find an effect in the task-dimension, but not in the distractor-dimension. However, without conducting an ANOVA we cannot know whether this prime congruency had different effects on Stroop and non-Stroop targets. What has been demonstrated is an effect of word-identity priming – something we shouldn't be too surprised to find. Experiment 4, a colour naming task, again examines early and late trials separately. In this experiment an effect of prime-target congruence is found in the later trials (here, the distractor-dimension of the Stroop task), but no attempt is made to test whether there was a differential effect on Stroop and non-Stroop trials.

Experiments 5 and 6 have modifications complementing those of experiments 3 and 4. The primes now have colour content, but no semantic content (the 'words' and always XXXX). The analysis presented are now based on the 'colour reading' and 'word reading' accounts presented at line 316 and following. It is not clear at line 354 whether the results being presented are from experiment 5 or from some combination of experiments 5 and 6 although all becomes clear a little later. The analyses are now made in terms of the congruence between the prime colour and target colour or between the prime colour and the target word. Evidence is found for an effect of congruence between prime colour

and target word but there is no evidence for an effect of congruence between prime colour and target colour. The task is word reading so the distractor dimension is colour. We see an effect of the distractor dimension but not the task dimension. Again, as no ANOVA is presented, we do not know if this effect modulates the Stroop effect.

In experiment 6 (colour naming) there were no effects of congruency between prime colour and target colour of word (even when trials were split into early and late blocks). The distractor dimension in a colour-naming task is word identify and, of course, that is not a variable in this experiment. Again, the analysis does not permit interactions between prime consistency and target (non) Stroop to be analysed.

The aim of all of these experiments has been to determine whether attention, as manipulated by the demands of the Stroop task, can affect 'high-level' visual information. The first two experiments certainly show that the primary task determines whether colour or word congruency between prime and target affects performance. In both cases only the distractor dimension of the prime affects the conscious Stroop effect. In the colour-naming (classic Stroop) task this effect only becomes evident once the task is well practiced. It is, of course, only through the meaning of the prime colour in the reverse-Stroop experiment (Expt 1) that an effect can be exerted. This seems good evidence that when an unseen stimulus is selected by attention the effect of its semantic properties on subsequent conscious decisions can be enhanced. I am not, however, sure whether, on this basis, one can claim that attention is operating on high-level unconscious visual information. It is much harder to make the claim that the effects of attention on word-congruence act at a semantic level. Congruence here implies identity between letters in prime and target and so the priming effect might operate at a level prior to reading (showing unconscious modulation of Stroop effects using colour-related words like 'sky', 'grass', or 'fire' would be much stronger evidence). I am not sure that the subsequent experiments added that much to the findings and I am extremely surprised that they were not analysed with ANOVAs so as to allow priming effects on the modulation of visible Stroop effects to be analysed. The authors' 'load theory' explanation of the development of priming over trials in colour naming tasks makes a great deal of sense. Unlike word reading, the colour naming task is demanding, and it is unsurprising that it is only affected by primes once it has become less practiced and requires less effort.

One concern I have is whether attention really is operating on high-level unconscious visual information in these experiments. Both revolve around the question of what it means for an item to be attended? Should an item be considered attended simply if attention enhances its processing or should only be thought of as attended if it is selected by attention (see e.g. Mole, 2008, *Journal of Consciousness Studies*, but see also Norman et al, 2013, *Psych Science*). In spatial attention we may attend to a region of space and mental resources may be allocated to items within that region of space, even though the items themselves were not selected. Are the items attended or is it only the space they occupy that is attended? Here, task demands are driving attention to colour or to words, and the processing of specific properties of the primes are being enhanced, but, in analogy with Mole's argument, one might claim that it is not blueness that is being selected when attention is directed towards colour in general. When processing of the unseen blue colour is enhanced it is enhanced by virtue of being a colour, so selection is not based on its semantics. Cueing conscious feature-based attention to blueness and then showing that semantic properties of an unseen blue stimulus (e.g. showing that as the unseen blue item was star shaped it primed responses to the visible word 'rocket' would surely qualify).

Even if we reject this argument out of hand, we might still worry whether attention really is operating on high-level unconscious visual information in these experiments. The blue coloured word (or XXXX non-word) is attended, but can we claim that it is attention that is operating on its semantic properties?

We know that the semantics of unseen items are processed (and indeed that semantic relatedness between unseen primes and visible target affects performance – van der Bussche et al, 2009, Experimental Psych) so is attention specifically responsible for semantic access here?

Overall I felt experiments 1 and 2 were interesting but I was not sure that the rest of the experiments were really necessary, especially when reported and analysed in a form that does not show how the primes are affecting the consciously attended and unattended dimensions of the target to which responses are made. It is clear that conscious task demands determine which properties of unseen primes affect behaviour. For colour primes it is also clear that these behavioural effects are mediated by the semantics of the attended unseen prime. I think more care has to be taken in arguing that attention is specifically acting on the unseen prime's semantic properties though.

BOB KENTRIDGE

Reviewer #3:

Remarks to the Author:

This very interesting study investigated the effect of task-induced attention (or maybe more accurately described as cognitive set) on the potential interference from invisible information. The authors devised an innovative double Stroop paradigm, with the first presentation rendered invisible using interocular suppression. Results convincingly showed a semantic level interference effect from the first stimulus that was modulated by the task as well as the attentional load on the second stimulus. The experiments are solid and the results are interesting. I would like the authors to consider and clarify the following issues (#3 is the more critical):

1. conceptually, the current study is quite similar with Kiefer & Martens (2010), even though the two studies were conducted with different methodologies. The authors cited this paper, but I believe it deserves more discussion, especially in terms the novel insights from the current study beyond the Kiefer & Martens study.
2. while the authors interpreted their results as effect from "task-induced attention", it seems a better description is cognitive set. At least it should be more clearly described as "feature" attention.
3. there is a potential confusion about where was the gating effect of attention/cognitive set applied. The title suggests that the authors believe the gating was on "unconscious semantic interference", but in the abstract and elsewhere, they seemed to suggest that their results "demonstrate the need of attention in extracting unconscious information". These two interpretations are fundamentally different. It is possible that EXTRACTING unconscious information is less dependent on attention/cognitive set, but the interference effect from the extracted information on the subsequent task is contingent on feature attention/set.
4. there is a lot of room for improvements in the writing. There are signs of sloppiness (e.g., repetition from lines 223-232), and many grammatical errors..

Reviewers' comments:

Reviewer #1 (Remarks to the Author):

Summary

Hung et al. evaluated the impact of unconscious, high-level processing with a novel task in which colored words were presented in a sequence where the prime stimulus was rendered invisible owing to interocular suppression. Participants either performed a word-reading task or a color-naming task for the visible word. Hung et al. found that the prime modulated the task performance in a task-specific manner. Specifically, suppressed word modulated the processing of the visible target word in word reading task but the impact of suppressed color was limited in the color-naming task.

Evaluation

It seems that the authors investigated an important question, to what extent the unconscious information is registered and how that is modulated by the task in relation to attention. Despite the small differences in key measures, the authors replicated the results over different stimulus conditions. I believe the claim would contribute to the literature in studies of attention and consciousness. However, it was really difficult to understand the manuscript. As a result, I was unable to evaluate the manuscript properly and I do not rule out a possibility that my concerns below could have been raised based on an incorrect understanding of the study.

1. Clarity of the manuscript

I found that the nature of the experiment was complex, where congruency of the prime and the target was independently manipulated, and the task, the word-reading, and color-naming, was varied across different experiments, and only a subset of conditions was pulled together for comparison. Despite its complexity, the manuscript was not clearly written and conditions were not clearly described, especially Exp 1 and 2. While the experimental conditions were straightforward to the authors, they are not for the readers.

In addition, some terms make the manuscript even more difficult to read. For example, What are Stroop and non-Stroop? It appears that the authors used “Stroop” when the color and word were incongruent, but I was not sure. Congruency and consistency were also mixed. Considering that congruency of the color and word was manipulated for the prime and the target independently, congruency can be applied to the prime, target, or the prime-target relation.

So, please make the manuscript easy to read and check out the PDF whether the figure was properly placed.

Response 1:

We apologize that the manuscript was not clear enough for the reviewer to assess our experimental design and analyses. As the reviewer had correctly inferred, Stroop/non-Stroop refers to the color-word congruence *within* the target word. The reviewer is absolutely correct in the notion of that congruency can be applied within the prime, the target, or in the prime-target relation. We chose prime-target color and word congruency as well as within target congruency (Stroop/non-Stroop) to analyze our data due to our hypotheses that responding to different aspects of target modulate how prime-target congruency affects target performance. We also apologize for using the words

congruency and consistency interchangeably. In the revision we stick to congruency for the relationship between prime and target. We have revised the manuscript thoroughly and clarified our experimental procedure, analyses, and results. The revised content is highlighted in blue.

Revision:

P.7. Line 8-13.

To examine whether prime-target word and color congruency affects target responses, we put in three different factors into our analysis: *prime-target color congruency*, *prime-target word congruency*, and *target word-color congruency* (that is, whether target is a Stroop word (word-color incongruent) or a non-Stroop word (word-color congruent)). In this article, we refer to prime-target color congruency as color congruency, prime-target word congruency as word congruency, and *target word-color congruency* as target Stroop/Non-Stroop throughout the manuscript.

2. Invisibility of the prime stimulus

In the literature, first, it is difficult to establish whether the invisible was really invisible because our judgment was susceptible to many factors including criterion and task demand (Yang et al., 2014). Second, distinguishing different types of invisibility matters (Kanai, Walsh & Tseng, 2010). This second issue is particularly important because attention interacts with interocular suppression in making the stimulus suppressed and, thus, its impact to unconscious processing depends on those two types of invisibility (Eo et al., 2016).

Eo, K., Cha, O., Chong, S. C., & Kang, M.-S. (2016). Less Is More: Semantic Information Survives Interocular Suppression When Attention Is Diverted. *The Journal of Neuroscience* :36(20), 5489–5497.

Kanai, R., Walsh, V., & Tseng, C.-H. (2010). Subjective discriminability of invisibility: A framework for distinguishing perceptual and attentional failures of awareness. *Consciousness and Cognition*, 19(4), 1045–1057.

Yang E, Brascamp J, Kang MS, Blake R (2014) On the use of continuous flash suppression for the study of visual processing outside of awareness. *Front Psychol* 5:724.

In Huang et al., the task difficulty changed between word reading and color naming. Participants performed the word-reading/color-naming task as well as the location judgment task, and they had to monitor whether the prime stimulus broke suppression whose location was uncertain. It is, therefore, possible that the prime could have been consciously registered but it was not reported in the experiment because the participants performed several tasks together and, thus, they could have adjusted their criterion more conservatively so that only the obvious, catch trials here, can be reported. Also, note that the previous studies have pointed out that information is registered when the invisibility is induced by the lack of attentional amplification.

It is, therefore, possible that the prime could have been consciously registered but it was not reported in the experiment because the participants performed several tasks together and, thus, they could have adjusted their criterion more conservatively so that only the obvious, catch trials here, can be reported.

Response 2:

First and foremost, we entirely agree with the reviewer that stimulus invisibility is always the upmost concern in any study that seeks unconscious effects. We also apologize that we did not make our experimental procedure clear enough to convince the reviewer that participants were indeed unconscious of the suppressed prime. We have revised our manuscript and added in more details to show how invisibility of the suppressed was assured. We also agree that a deeper issue is how the stimulus invisibility was achieved as the previous studies (Eo et al., 2016; Kanai, et al., 2010) pointed out that attention can substantially interact with visual suppression/masking and influence the stimulus invisibility. We have added in a short discussion on this point. A direct examination on how attention contributes to stimulus invisibility in our paradigm will require future studies.

In each trial, participants had three tasks, conducted separately in three different phases of the experiment in sequence. All completed via button press. (1) During the suppression period, a detection task is required. They were instructed to respond as soon as any part of the word stimulus had been detected, which prevented factors such as forgetting, attention-divergence, interference, or task load. It was stressed that this is the most important task and even seeing a stroke of the word counted as detection. If the prime was detected, the trial ended immediately. (2) If the prime was not detected, the trial continued to the visible target period, and the participant had to name the color or the word of the visible target word, depending on which experiment they were in. It was stressed that both speed and accuracy were important. (3) At the end of the trial, a two-alternative-force-choice (2AFC) location task asked the participant to judge the location of the suppressed prime. It was made clear that they should try their best to judge even if the prime remained undetected in the suppression period. Please note that this 2AFC location task did not require the participant to hold prime information until the end of the trial but instead served as a reconfirmation of prime invisibility. Thus, the prime invisibility was assessed with both an immediate subjective and a retrospective objective criteria.

We believe that our interocular suppression was successful for three reasons. First of all, participants' near-ceiling accuracy on the visible catch trials showed that they were paying attention to the detection task and tried their best to detect the prime. Secondly, we also calibrated the prime luminance according to their detection on the suppressed prime with a staircase procedure on a trial-by-trial basis. Using a 3-up-1-down calibration staircase: when a suppressed stimulus was detected, the contrast decreased in the next trial, while if a suppressed stimulus was not detected three trials in a row, the contrast increased in the next trial. The group average breakthrough rates on the suppressed prime in Experiments 1~6 were 25.13 (1.85)%, 25.19 (1.62)%, 29.16 (3.55)%, 23.96 (1.00)%, 26.16 (2.22)%, and 23.87 (1.41)%. These numbers suggest that the calibration was successful, and across participants there was little individual differences. If some participant had been conservative and neglected primes that he/she had actually detected, we would expect the calibration to be a failure. For instance, the participant might break suppression rarely even with our online adjustment of the prime luminance. To err on the safe side, as we could not tell whether the participants were simply conservative or really did not perceive the stimulus, these participants had been excluded prior to analysis. Thirdly, their chance performance on the 2AFC prime location task objectively indicated the invisibility of the suppressed prime.

Furthermore, although at the group level, participants' performance on the location task was not different from chance, it could very well be that the results were driven by a few participants that exhibited higher than chance rate performance on the 2AFC location task. To address this

possibility, we went on to perform a binomial test (one-tailed, $p > .05$) on each participant on their location task performance and re-analyzed the data with only those participants with chance performance (Lin & Murray, 2014). The data in all experiments show similar patterns. In Experiment 1, 17 out of 20 participants performed at chance. we again performed three-way repeated measure ANOVA (*prime-target color congruency*, *prime-target word congruency*, and *target word-color congruency*). The main effect of target (reverse) Stroop was found, $F(1, 16) = 20.38, p = .0004, \eta_p^2 = .56$. Furthermore, there was a marginal interaction between color and word consistency, $F(1, 16) = 3.97, p = .06, \eta_p^2 = .20$ as well as color and target Stroop, $F(1, 16) = 6.51, p = .02, \eta_p^2 = .29$. In Experiment 2, the same analysis was performed. The main effect of target Stroop was found, $F(1, 16) = 41.00, p = .00000, \eta_p^2 = .73$, and no further effect from the suppressed prime was found (word: $F(1, 16) = 1.05, p = .32, \eta_p^2 = .07$; color: $F(1, 19) = 0.04, p = .85, \eta_p^2 = .00$). Identical analysis performed on the 1st Quarter trials showed very similar pattern to the overall data: The main effect of target Stroop was found, $F(1, 16) = 46.07, p = .00000, \eta_p^2 = .75$, with no further effect from the prime-target relationship (word: $F(1, 16) = 2.34, p = .15, \eta_p^2 = .14$; color: $F(1, 19) = 0.1, p = .75, \eta_p^2 = .01$). However, the result on the 4th Quarter trials showed a marginal main effect of word incongruency effect $F(1, 16) = 3.87, p = .07, \eta_p^2 = .21$, but not color incongruency effect $F(1, 19) = 0.18, p = .68, \eta_p^2 = .01$, in addition to the main effect of target Stroop, $F(1, 16) = 15.10, p = .002, \eta_p^2 = .50$. In Experiment 3, all participants performed at chance. In Experiment 4, 16 out of the 20 participants performed at chance. The word-induced semantic incongruency effect was marginally significant with $t(19) = -1.93, p = .07$, Cohen's $d_{av} = 0.52$, congruent trials: 0.3% slower; incongruent trials: 7.1% slower. In Experiment 5, 19 out of 20 participants performed at chance. The color-induced semantic incongruency effect was significant with $t(19) = -2.27, p = .04$, Cohen's $d_{av} = 0.58$, semantic congruent trials: 2.8% slower; semantic incongruent trials: 6.5% slower. In Experiment 6, the color-induced semantic incongruency effect remained insignificant, either in all trials ($t(19) = -0.73, p = .48$, Cohen's $d_{av} = 0.22$, congruent trials: 3.1% slower; incongruent trials: 4.5% slower) or later trials ($t(19) = -0.58, p = .57$, Cohen's $d_{av} = 0.19$, congruent trials: 3.6% slower; incongruent trials: 6.3% slower). In Experiment 7, all participants performed at chance. In Experiment 8, 18 out of 20 participants performed at chance at the location task. The semantic incongruency effect again interacted with the location, showing the effect only when target and prime were co-localized. A direct comparison on the co-localized trials yielded similar significant results ($t(19) = -2.3, p = 0.03$, Cohen's $d_{av} = 0.63$, congruent trials: 1.3% slower; incongruent trials: 5.7% slower).

We agree that any of the single criterion was not sufficient to claim that participants were unconscious of the prime, and measuring stimulus visibility always has its unavoidable uncertainty. However, all above converging evidence elicited converging evidence of the prime invisibility. We have revised our manuscript accordingly and added in one section in the supplementary information *Invisibility of the suppressed prime* to clarify how we ensured the invisibility of the suppressed stimulus.

Revision:

Main text

Method

P.29. Line 5-9.

Each color had its own 3-up-1-down contrast calibration staircase: when a suppressed stimulus was detected, the contrast decreased in the next trial, while if a suppressed stimulus was not detected three trials in a row, the contrast increased in the next trial. The initial contrast was chosen according to participants' performance on the 20 practice trials before they proceeded the actual experiment.

P.29 Line 21-29. P.30. Line 1-4.

In each trial, a participant had three tasks in sequence. All completed via button press. (1) During the suppression period, a detection task is required. He/She was instructed to respond as soon as any part of the word stimulus had been detected. It was stressed that this is the most important task and even seeing a stroke of the word counted as detection. If the prime had been detected, the trial ended immediately. (2) If the prime was not detected, the participant had to name the color or the word of the visible target word, depending on which experiment they were in. It was stressed that both speed and accuracy were important. (3) At the end of the trial, a two-alternative-force-choice (2AFC) location task asked the participant to judge the location of the suppressed prime. It was made clear that they should try their best to judge even if the prime remained undetected in the suppression period. Please note that this 2AFC location task did not require the participant to hold prime information until the end of the trial but instead served as a reconfirmation of prime invisibility. Thus, the prime invisibility was assessed with both an immediate subjective and a retrospective objective criteria.

Discussion

P.26. Line 12-27.

As our study showed that attention load of the task modulated the semantic interference from an unconscious prime, it is important to point out that previous studies have shown that attention also interacts with how stimulus invisibility is achieved under a visual suppression/masking paradigm (Kanai et al., 2010; Eo et al., 2016). For example, it has been shown that semantic information is registered when the invisibility is induced by the lack of attention under interocular suppression (Eo et al., 2016). How exactly did the task-induced attention load, and any other factors related to task, interacted with interocular suppression in our study will require future studies. However, to ensure that stimulus invisibility was properly achieved in our experiments, we have proceeded to perform additional analyses on our objective awareness test (i.e. 2AFC prime location task). Apart from having near-ceiling performance on visible and blank catch trials as well as group-level chance performance on the 2AFC location task, we ran a binomial test on the responses of the 2AFC location task on every single participant and analyzed our data on those participants that showed chance performance (Lin & Murray, 2014). The results remained largely identical (see supplementary information, *invisibility of the suppressed prime*). Whereas measuring stimulus visibility always has its unavoidable uncertainty, we believe our results showed converging, realistically-achievable evidence of the prime invisibility.

Supplementary information

2. Invisibility of the suppressed prime

We ensured the invisibility of the suppressed time at two levels. Firstly, we looked at group-level performance. The interocular suppression was successful in making the prime invisible for the following reasons. First of all, participants' near-ceiling accuracy on the visible catch trials showed that they were paying attention to the detection task under suppression (Figure S1, Bottom, light gray bars). Secondly, during the suppression period, we also calibrated the prime luminance according to their detection on the suppressed prime with a staircase procedure. If some participant had been conservative and neglected primes that he/she has actually detected, we would expect the calibration to be a failure. Using a 3-up-1-down calibration staircase, the group average breakthrough rates on the suppressed prime in Experiments 1~6 were 25.13 (1.85)%, 25.19 (1.62)%, 29.16 (3.55)%, 23.96 (1.00)%, 26.16 (2.22)%, and 23.87 (1.41)% (Figure S1, Top left). According to the 3-up-1-down staircase, we expected to see the final thresholds converge around 25% of the full contrast. These numbers suggest that the calibration was successful, and across participants there was little individual differences. Thirdly, their chance performance (i.e. 50%, above or below fixation) on the 2AFC prime location task objectively indicated the invisibility of the suppressed prime (Figure S1, Top right).

Secondly, we examined in all experiments if each participant performed at chance level on the 2AFC prime location localization using a binomial test (one-tailed, $p > .05$) and re-analyzed the data with only those participants with chance performance (Lin & Murray, 2014). In Experiment 1, 17 out of 20 participants performed at chance. we again performed three-way repeated measure ANOVA (*prime-target color consistency*, *prime-target word consistency*, and *target word-color consistency*). The main effect of target (reverse) Stroop was found, $F(1, 16) = 20.38$, $p = .0004$, $\eta_p^2 = .56$. Furthermore, there was a marginal interaction between color and word consistency, $F(1, 16) = 3.97$, $p = .06$, $\eta_p^2 = .20$ as well as color and target Stroop, $F(1, 16) = 6.51$, $p = .02$, $\eta_p^2 = .29$. In Experiment 2, the same analysis was performed. The main effect of target Stroop was found, $F(1, 16) = 41.00$, $p = .00000$, $\eta_p^2 = .73$, and no further effect from the suppressed prime was found (word: $F(1, 16) = 1.05$, $p = .32$, $\eta_p^2 = .07$; color: $F(1, 19) = 0.04$, $p = .85$, $\eta_p^2 = .00$). Identical analysis performed on the 1st Quarter trials showed very similar pattern to the overall data: The main effect of target Stroop was found, $F(1, 16) = 46.07$, $p = .00000$, $\eta_p^2 = .75$, with no further effect from the distractor-target relationship (word: $F(1, 16) = 2.34$, $p = .15$, $\eta_p^2 = .14$; color: $F(1, 19) = 0.1$, $p = .75$, $\eta_p^2 = .01$). However, the result on the 4th Quarter trials showed a marginal main effect of semantic incongruency effect $F(1, 16) = 3.87$, $p = .07$, $\eta_p^2 = .21$, but not color incongruency effect $F(1, 19) = 0.18$, $p = .68$, $\eta_p^2 = .01$, in addition to the effect of target Stroop, $F(1, 16) = 15.10$, $p = .002$, $\eta_p^2 = .50$. In Experiment 3, all participants performed at chance. In Experiment 4, 16 out of the 20 participants performed at chance. The word-induced semantic incongruency effect was marginally significant with $t(19) = -1.93$, $p = .07$, Cohen's $d_{av} = 0.52$, semantic congruent trials: 0.3% slower; semantic incongruent trials: 7.1% slower. In Experiment 5, 19 out of 20 participants performed at chance. The color-induced semantic incongruency effect was significant with $t(19) = -2.27$, $p = .04$, Cohen's $d_{av} = 0.58$, semantic congruent trials: 2.8% slower; semantic incongruent trials: 6.5% slower. In Experiment 6, the color-induced semantic

incongruency effect remained insignificant, either in all trials ($t(19) = -0.73, p = .48$, Cohen's $d_{av} = 0.22$, semantic congruent trials: 3.1% slower; semantic incongruent trials: 4.5% slower) or later trials ($t(19) = -0.58, p = .57$, Cohen's $d_{av} = 0.19$, semantic congruent trials: 3.6% slower; semantic incongruent trials: 6.3% slower). In Experiment 7, all participants performed at chance. In Experiment 8, 18 out of 20 participants performed at chance at the location task. The semantic incongruency effect again interacted with the location, showing the effect only when target and prime were co-localized. A direct comparison on the co-localized trials yielded similar significant results ($t(19) = -2.3, p = 0.03$, Cohen's $d_{av} = 0.63$, semantic congruent trials: 1.3% slower; semantic incongruent trials: 5.7% slower).

Overall, our data showed that at the group level, participants paid attention to the detection task and responded accordingly, reflected by successful calibration results and near-ceiling performance on both blank and visible trials. Moreover, even when participants who performed above chance level on the 2AFC prime location task were excluded, the results showed similar patterns as we found in all participants. We agree that any of the single criterion was not sufficient to claim that participants were unconscious of the prime, and measuring stimulus visibility always has its unavoidable uncertainty. However, all above results elicited converging evidence of the prime invisibility.

Figure S1. **Top left.** The breakthrough rates of the suppressed prime in all experiments. **Top right.** 2AFC Prime location task accuracy in all experiments. **Bottom.** Accuracy on visible and blank catch trials in all experiments. Each bar represents group results (n=20) from each experiment. Error bars denote stand error of the mean (SEM).

3. Theoretically,

Attention can interact with consciousness. I do not have any problem with this claim. However, it is difficult to take the claim for several reasons.

First, Hung et al. did not directly manipulate the load. Instead, they varied the task. If task changes, task set should change accordingly throughout the experiment. The changed task set can influence several aspects of the experiment.

For example, in the word-naming, it is possible that the participants looked into a set of orthographical features of RED and BLUE, instead of processing word RED and BLUE, because there were only two colors and two words. Considering that there was converging evidence that low-level features are registered for the suppressed stimulus, the observed effect could be due to the feature processing rather than high-level semantic processing.

Response 3:

We thank the reviewer to point this out. In fact, there are two important points embedded in this comment. Firstly, does our attention modulation come from task load or task set differences? Secondly, what exactly was modulated by attention in our experiments? Each of the two questions cannot be simply answered by the first six experiments in that (1) as the reviewer pointed out, task load and task set manipulation were intertwined in these experiments. Word-naming, throughout these experiments, was of lower task load and oriented participants to the semantic feature of the target. On the other hand, color-naming was of higher task load and prompted attention to be directed to color feature of the target. As we are focusing on the semantic interference from the unconscious prime, word-naming will be considered with aligned task set while color-naming will be with unaligned task set. (2) Since the primes and targets had identical word forms, the attention modulation could operate on the orthographic or the semantic relationship between the prime and target, or both.

In order to conclusively show that semantic congruency can be modulated by how participants distribute their attention in the task, we performed two more experiments. In Experiments 7 & 8, we chose two new lowercase words as the visible targets “scarlet” and “navy” in blue or red (scarlet, navy) while the suppressed primes remains colorless “BLUE” or “RED” so that these words still exhibited a semantic but not orthographic (dis)similarity **with the prime**. Importantly, the change of word frequency reverses the relationship between the task and load, word-naming was still accompanied by an interference from the target color (reverse Stroop effect), while the typical Stroop effect from word semantic interference during color-naming disappeared. This opposite pattern of the strength of Stroop/reverse Stroop effects was later confirmed with twenty participants in each experiment. With strong reverse Stroop effect and no Stroop effect on the targets, our new experiments disentangled the task load/task set relationship. Word-naming is now characterized as high load (due to the reverse Stroop effect) yet with aligned task set, while color-naming is now low load (due to the disappearance of Stroop effect) with unaligned task set. Two separate pairs of conflicting hypotheses can be articulated on (1) the modulated feature:

orthography or semantics and (2) the modulating source: task load or task set. In (1), if pure semantics from an unconscious prime could exert an effect, we expected to see unconscious interference on the target responses. Otherwise if the interference effect was mainly driven by orthographic similarities, the effect will disappear. In (2) if task load played a key role in modulating the interference, the interference would appear in the new low load experiment (Exp 8, color-naming) but not the new high load experiment (Exp 7, word-naming, or only in the later practiced trials). Otherwise if task set alignment played a key role, the interference would appear in the word naming experiment (Exp 7), but not the color naming experiment (Exp 8).

Our results showed that interference between prime and target still persisted, suggesting that pure semantics between prime-target could exert an effect. Moreover, this semantic interference only occurred in the low load (Exp 8, color naming) but not high load experiment (Exp 7, word naming), suggesting that task load, but not task set alignment, was the determining modulator.

Here we present the numerical results. Firstly, we observed opposite patterns of Stroop/reverse-Stroop effects. In Experiment 7 when the participants were instructed to name the word, a significant reverse Stroop effect arose ($t(19) = 3.15$, $p = 0.005$, Cohen's $d_{av} = 0.32$), while in Experiment 8, during color-naming, the typically stronger Stroop effect completely disappeared ($t(19) = 0.85$, $p = 0.41$, Cohen's $d_{av} = 0.09$). This showed that word-naming experiment was of high load while color-naming was of low load, which was the opposite of typical Stroop/reverse Stroop effects.

In Experiment 7, the semantic congruency effect was neither significant in all trials ($t(19) = 0.82$, $p = .42$, Cohen's $d_{av} = 0.19$) nor the later trials ($t(19) = 1.21$, $p = 0.24$, Cohen's $d_{av} = 0.33$). On the other hand, such effect was significant in Experiment 8 in all trials with an interaction with prime location. That is, only when the invisible prime and visible target were in the same side of fixation point (locations were still jittered), an invisible prime (e.g. BLUE) prompted a slower response to a subsequent visible semantically incongruent target (e.g. Scarlet), as compared to a congruent one (e.g. Navy).

How do these two new experiments shed light on our interpretation of the previous findings? First of all, we show that even when orthographic (word-form) similarity was removed between a unconscious prime and a conscious target, semantic incongruency between them exerted a slowing interference effect on target response. Such result suggests a pure conceptual/semantic relationship could be processed under interocular suppression. Furthermore, this interference was modulated by *task load*. When task load is low (color-naming, no Stroop effect), the interference arises. On the contrary, when task load is high (word-naming, reverse Stroop effect), the unconscious interference disappears. We could thus pinpoint that task load, but not task set alignment, played a key role in modulating unconscious semantic interference.

We have added in the new experiments, detailed analyses, and discussions into the manuscript.

Revision:

Introduction

P.5. Line 21-29.

..... Meanwhile, responding to different aspects of a target word could selectively activate feature-specific attention, which could be further utilized to select the subliminal relevant feature. That is, responding to the semantic feature of a supraliminal word could selectively deploy semantic-specific attention to the subliminal semantic feature. Two hypotheses can thus be raised and examined. (1) An unconscious effect from high-level visual content requires *general attentional resources*. That is, an unconscious effect will occur only when the task load is low and additional attentional resources could be distributed to the unconscious stimulus. (2) An unconscious effect from high-level visual content requires *feature-specific attention*. That is, an unconscious effect will occur only when the task set is tuned to the critical subliminal feature.

Results

P. 17-20 and table on P.21

Attention modulation on unconscious semantic interference: task load or task set? semantic congruency or orthographic similarity?

In Experiments 1~4, we found that when the subliminal prime and supraliminal target were semantically incongruent (e.g. BLUE vs. RED), the incongruency prompted slower target responses. This effect was further shown to be modulated by task load. When the task load was low (Experiments 1 & 3, word naming), the effect appeared in all trials. When the task load was high (Experiments 2 & 4, color naming), the effect appeared only in the later trials when there was a practice effect on the task. This effect was interpreted as a word-induced subliminal semantic interference. However, as the subliminal primes and targets had identical visual forms (BLUE/RED), one may argue that the effect could simply emerge from low-level visual feature adaptation. Although in all experiments, the prime and target were of different font sizes and presented on jittered locations. We first proceeded to examine if prime-target co-localization (on the same or different side of the fixation point) interacted with the word congruency effect with the reasoning that if stronger effect arose when the prime and target were spatially closer, low-level visual adaptation may contribute significantly to our word incongruency effect. We thus ran an additional two-way (word congruency; prime-target location consistency) repeated measures analysis of variance on the data of Experiments 3 & 4. The results from both experiments showed no interaction between location and semantic congruency (Experiment 3, all trials, $F(1, 19) = 0.41$, $p = 0.53$, $\eta_p^2 = .004$; Experiment 4, 4th quarter trials, $F(1, 19) = 0.48$, $p = 0.50$, $\eta_p^2 = .01$). Such results suggested that our findings that prime-target word incongruency slowed down target responses were not merely driven by low-level adaptation of visual stimulus shapes.

However, since our unconscious primes and conscious targets shared not only semantic but also orthographic similarities, it is almost impossible to tease apart if such effect was due to word form (orthographic) incongruency or semantic/conceptual incongruency, or both. To directly examine whether task-induced attention could modulate pure semantic interference, we adopted the design in Experiments 3 and 4 with one adjustment: replacing the target words to “scarlet” and “navy.” By utilizing these two target words, the prime and target now contained no orthographic similarities while still exhibited semantic/conceptual (in)congruencies. It has been shown that once the color association between colors and words is weaker (e.g. naming the color blue on the word *red* vs. *and*), the Stroop interference from color-naming weakens²⁰. The authors also proposed that word frequency could play a key role as low word frequency is typically linked to weaker association with colors. This early observation led to the prediction in our current study that color-

naming on the words *navy* and *scarlet* would accompany weaker word semantic interference (i.e. weaker Stroop effect) as demonstrated in the original study. Indeed, in our pilot data ($n = 2$), this choice of lower-frequency words as targets reversed the relative task load for word naming and color naming: Word-naming was still accompanied by an interference from the color of the target (reverse Stroop effect), while the typical Stroop effect from color-naming disappeared. This opposite pattern of the strength of Stroop/reverse Stroop effects was later confirmed with twenty participants in each experiment. Such pattern teased apart the tight relationship between task load and task set alignment in our previous experiments. That is, the stronger unconscious prime semantic incongruity effect in word-naming experiments can be attributed to either lower task load or task set alignment (attending to semantic feature). While weaker semantic incongruity effect in color-naming can be due to high task load or task set unalignment (attending to non-semantic feature). Thus, it is again almost impossible to pinpoint whether task-induced attentional load or feature-specific attention modulated the unconscious effect. Nonetheless, with the current design, we have created a high-load x task-effect aligned feature-attending (word naming) and low-load x task-effect unaligned feature-attending (color-naming). The new experiments allowed us to examine two separate pairs of conflicting hypotheses on (1) the modulated feature: orthography or semantics and (2) the modulating source: task load or task set. In (1), if pure semantics from an unconscious prime could exert an effect, we expected to see unconscious interference on the target responses. Otherwise if the interference effect was mainly driven by orthographic similarities, the effect would disappear. In (2), if task load played a key role in modulating the interference, the interference would appear in the new low load experiment (Experiment 8, color-naming) but not the new high load experiment (Experiment 7, word-naming, or only in the later practiced trials). Otherwise if task set alignment played a key role, the interference would appear in the word-naming but not the color-naming experiment.

Firstly, we again obtained similar data on successful interocular suppression in both experiments (Experiment 7: location task accuracy 49.81% (1.15%), paired $t(19) = -0.16, p = 0.87$; Experiment 8: 50.2 % (1.33 %), paired $t(19) = 0.15, p = 0.88$). Moreover, the near-ceiling performance was found on the blank and visible trials with 97.78% (0.51%) and 98.12% (1.29%) accuracies in Experiment 7 and 97.81% (0.54%) and 99.69% (0.31%) accuracies in Experiment 8. The critical difference was that the reverse Stroop effect was evident in Experiment 7 (Reaction time comparison of Stroop and non-Stroop trials: $t(19) = 3.15, p = 0.005$, Cohen's $d_{av} = 0.32$) while the Stroop effect was not in Experiment 8 (Reaction time comparison of Stroop and non-Stroop trials: $t(19) = 0.85, p = 0.41$, Cohen's $d_{av} = 0.09$).

In Experiment 7, the RTs in each condition were first normalized against blank trials. A direct paired comparison between semantically congruent and incongruent trials showed null effect ($t(19) = 0.82, p = .42$, Cohen's $d_{av} = 0.19$, Fig. 6A). On average, compared to the blank trials, semantic congruent trials were 2.1 % slower while incongruent ones 0.9 % were slower. We also examined whether prime-target location (same-different) had interacted with the semantic congruency effect. A two-way (semantic congruency; prime-target location consistency) repeated measures analysis of variance was performed on the normalized target word reaction time. Neither the main effect of semantic congruency, $F(1, 19) = 0.09, p = 0.77, \eta_p^2 = .005$, nor the main effect of location congruency was found, $F(1, 19) = 1.32, p = 0.26, \eta_p^2 = .07$. There was no interaction between the two, $F(1, 19) = 0.21, p = 0.65, \eta_p^2 = .01$. Similar results were found in the 4th quarter trials (direct comparison: $t(19) = 1.21, p = .24$, Cohen's $d_{av} = 0.33$; semantic congruency x location consistency

2-way ANOVA: main effect of semantic congruency $F(1, 19) = 0.07, p = 0.80, \eta_p^2 = .004$; main effect of location: $F(1, 19) = 0.11, p = 0.75, \eta_p^2 = .005$; interaction: $F(1, 19) = 0.49, p = 0.49, \eta_p^2 = .03$.)

In Experiment 8. A direct paired comparison on the normalized RT between semantically congruent and incongruent trials showed null effect ($t(19) = -1.12, p = .28$, Cohen's $d_{av} = 0.24$). On average, compared to the blank trials, semantic congruent trials were 1.32 % slower while incongruent ones 2.7 % were slower. However, a two-way (semantic consistency; prime-target location consistency) repeated measures analysis of variance showed an interaction between the prime-target location and semantic congruency with $F(1, 19) = 8.13, p = 0.01, \eta_p^2 = 0.30$. The main effects were not significant (semantic congruency: $F(1, 19) = 1.1, p = 0.29, \eta_p^2 = 0.06$; location congruency: $F(1, 19) = 0.05, p = 0.83, \eta_p^2 = 0.003$). A planned post hoc comparison between semantic congruent and incongruent trials when the prime-target were co-localized showed a significant effect with $t(19) = -2.45, p = .02$, Cohen's $d_{av} = 0.63$, Fig. 6B).

Fig. 6. Experiments 7 and 8 results. Y-axis denotes reaction time, and X-axis denotes two critical conditions: semantically congruent (SEM-CON) and semantically incongruent (SEM-INCON). Black dots and lines denote longer RT in the semantically incongruent condition while gray dots and lines denote longer RT in the semantically congruent condition. Each pair of dots lined between the two conditions represents one participant. Upward lines indicate longer RT increase in the incongruent condition while downward lines indicated RT decrease in the incongruent condition. The bars denote group mean with the error bars indicating standard error of the mean (SEM). Asterisk denotes significance. Please note that the data from Experiment 8 was from prime-target co-localized trials.

These results showed that, without orthographic similarity, the semantic incongruency between the prime and target did pose a slowing effect on the target responses if that they were co-localized. The observation that the semantic effect further interacted with prime-target co-localization indicated the need of spatial attention toward the primed location for such effect to occur. This semantic effect was also modulated by the task load. In Experiment 8 (color-naming), when the task load was low as indicated by the lack of the Stroop effect, the semantic incongruency slowed down target response. On the contrary, in Experiment 7 (word-naming), when the task load was high as indicated by the reverse Stroop effect, such effect disappeared. As task alignment and task load were separated in these two experiments (i.e. Experiment 7: Semantic task but high load, Experiment 8: Non-semantic task but low-load), distinct from all previous experiments. This particular finding stressed that the level of task load, but not task alignment, modulated an unconscious semantic effect in our paradigm. Table 1 summarizes the stimulus type, task, and effect in all experiments.

Experiment	Prime	Target	Task	Load	Interference
1	BLUE BLUE RED RED		Word-naming	Low	Yes
2	BLUE BLUE RED RED		Color-naming	High	Yes after practice
3	BLUE RED	BLUE BLUE RED RED	Word-naming	Low	Yes
4	BLUE RED	BLUE BLUE RED RED	Color-naming	High	Yes after practice
5	XXXX XXXX		Word-naming	Low	Yes
6	XXXX XXXX		Color-naming	High	No
7	BLUE RED	navy navy scarlet scarlet	Word-naming	High	No
8	BLUE RED	navy navy scarlet scarlet	Color-naming	Low	Yes

Table 1. The table summarizes all eight experiments where we manipulated the color, word, and semantic relationship between the subliminal prime and supraliminal target. Task (word-naming vs. color-naming) and task load (high vs. low, according to the strength of Stroop/Reverse Stroop effect) in each experiment are presented.

Discussion

P.22 Line 8-32. P.23. Line 1-7.

Two important questions were left answered after our first six experiments. Firstly, although the general predictions of the load theory are compatible with our findings, task-induced attention may *selectively* trigger different cognitive sets under different tasks, which in turn gates how an unconscious prime interfered with subsequent target performance. For example, in the word-naming experiments (Experiments 1, 3, & 5), participants' attention was selectively oriented to the word semantics. Such attention tuning would have been applied further to the suppressed prime and allowed semantic interferences to occur. On the contrary, in the color-naming experiments, attention was deployed to the word color, which would have interfered the semantic processing of the target (i.e. the Stroop effect) as well as hindered the semantic interferences from the suppressed prime. Since both our interfering effects were semantic in nature, word- and color-induced semantic congruency effects were gated by this cognitive-set-induced-tuning to different levels: Under word-naming task, both exerted an effect; while under color-naming task, only the more automatic word-induced incongruency effect re-emerged after the participants became more fluent in the task (i.e. 4th quarter trials in Experiments 2 & 4). Secondly, one could argue that since identical word-forms were used as primes and targets, what appeared to be a semantic incongruency effect could be driven merely by orthographic dissimilarity. Although we had jittered the prime-target locations and font sizes, and our further analyses focusing on the interaction between the semantic effect and location yielded null effect, which excluded the possibility of low-level adaptation.

Thus, (1) to ensure that sheer semantic effect could be modulated and (2) to tease apart cognitive-set-induced-tuning and task load modulation, in Experiments 7 and 8, we went one step further to remove the orthographic similarity of the prime and target while maintaining the semantic relationship by introducing a different set of targets: *navy* & *scarlet*. Importantly, the Stroop but not the reverse Stroop effect disappeared when participants were instructed to name the color and word, respectively. Such manipulation thus enabled a misalignment between cognitive-set-

induced-tuning attention and task load, allowing us to pinpoint the determining factor in modulating the unconscious effect. As a result, we showed the unconscious semantic incongruency effect when the task load was low yet non-semantic (Experiment 8, color-naming) but not when the task load was high but semantic (Experiment 7, word-naming), suggesting that general availability of attentional resources, but not cognitive-set-induced-tuning attention, played a key role in modulating the unconscious semantic interference in our study.

Second, I am not still sure why the suppressed word can slow down the target word processing. The effect was very small as well. One way to demonstrate the presence of the effect was to do a task where both the prime and the target are visible. I believe establishing the double Stroop procedure with the visible stimuli would be important in building an argument, especially for discussing the cognitive processes involved in the task-specific priming effect as well as for providing some baseline level performance we can expect from the procedure.

Response 4:

An early study (McClain, 1983) has reported that semantic congruency between a conscious prime word and a subsequent colored word speeds up the color-naming response of the colored word. In one condition participants were instructed to name the colored word *red*, *blue*, *green*, and *yellow* in an incongruent color (e.g. *red* in color blue). The result showed that participants responded faster when the colored words were preceded by semantically related primes (the same words in black ink) than when the colored words were preceded by irrelevant primes. Moreover, generally speaking, subliminal words have been shown repeatedly to slow down the response of a subsequent incongruent word, as compared to a congruent one. For example, Dehaene et al. (1998) showed that in a number classification task (judging whether a number was larger or smaller than 5), a masked prime, if incongruent with the later number digit, led to slower response time. More recently, Costello et al. (2009) showed that after a visible prime, semantically related suppressed target words broke through interocular suppression faster, as compare to the unrelated ones. These studies have established that semantic incongruency/unrelatedness between a conscious and an unconscious word typically slowed down responses.

McClain, L. (1983). Color priming affects Stroop interference. *Perceptual and Motor Skills*, 56(2), 643-651.

Dehaene, S., Naccache, L., Le Clec'H, G., Koechlin, E., Mueller, M., Dehaene-Lambertz, G., ... & Le Bihan, D. (1998). Imaging unconscious semantic priming. *Nature*, 395(6702), 597.

Costello, P., Jiang, Y., Baartman, B., McGlennen, K., & He, S. (2009). Semantic and subword priming during binocular suppression. *Consciousness and cognition*, 18(2), 375-382.

We also find this comment relevant to the subsequent comment. As the reviewer pointed out, another expectation of our results is negative priming, which we did not find. Please see more discussion below.

Third, in a related vein, negative priming has been extensively discussed in the Stroop literature. I think the effect reported in the study seems to have some relevance to this literature. Authors may find some insight or information from the literature.

Response 5:

We appreciate this comment. Indeed, negative priming has been a great tool in the Stroop literature to examine different processes. We have taken this chance to add in a paragraph in the supplementary information to discuss previous findings on negative priming in the Stroop paradigm and how they relate to our current findings.

Revision

Supplementary information

3. Negative priming in the Stroop paradigm

Our color-naming experiment (e.g. Experiment 2) showed that in the later trials when there was a significant practice effect on the target responses, incongruent prime-target words led to slower responses. This was different from the typical negative priming effect observed in the Stroop literature (Neil, 1977; Besner, 2001; Mari-Beffa, Estévez, Danziger, 2000). In the context of the Stroop paradigm, negative priming refers to the phenomenon where a prime and target being related in an ignored stimulus dimension leads to interferences (e.g. slower responses). For example, Mari-Beffa et al. (2000) examined that when the prime did not elicit the Stroop effect, was such null result sufficient to conclude that word processing did not occur, and thus word-reading was not automatic. They approached this question with a prime-target negative priming paradigm where after the prime, participants had to name of color of a target, which was always color-word inconsistent. They found that albeit no Stroop effect was found on the response to prime, there was a clear negative priming from the prime to the target. That is, when the two words were related, people responded *slower* to the target. This finding is different from the negative interference effect we report here in that we show when the prime-target words were incongruent, people responded slower. One major difference is that in their design, the target was always color-word inconsistent. In order to correctly respond to the target, participants had to always suppress the word processing, and a related prime-target relationship made the suppression more difficult, leading to the negative priming. Such negative priming effect could indeed be suitable to be combined with our paradigm to further examine what exactly can be extracted and processed under interocular suppression.

Neill, W. T. (1977). Inhibitory and facilitatory processes in selective attention. *Journal of Experimental Psychology: Human Perception and Performance*, 3(3), 444-450.

Besner, D. (2001). The myth of ballistic processing: Evidence from Stroop's paradigm. *Psychonomic Bulletin & Review*, 8(2), 324-330.

Mari-Beffa, P., Estévez, A. F., & Danziger, S. (2000). Stroop interference and negative priming: Problems with inferences from null results. *Psychonomic Bulletin & Review*, 7(3), 499-503.

4. Other concerns and questions.

- How did the authors calculate the accuracy of the blank condition? The task was a 2AFC location judgment task instead of a presence/absence judgment.

Response 6:

The accuracy of the catch trials was calculated from their detection task performance, not the location task. That is, if they successfully detected on the visible catch trial, that was a hit; similarly, no response during the blank catch trial was considered a correct rejection. We have clarified accuracy calculation in the manuscript. Please also refer to our revised Fig. 1.

Revision:

P.7. Line 4-6.

Moreover, the performance on the blank and visible trials *on the detection task* was 97.81% (1.04%) and 98.44% (0.77%), respectively, showing high accuracy and consistency.

Fig. 1. Trial sequence, stimulus, and task. Each trial was self-paced and began with a varied SOA ranging from 0.1~1 s. After which a dynamic flashing Mondrian pattern was presented to the dominant eye while the colored word prime was presented to the non-dominant eye. During 400 ms suppression period, the suppressed word was sandwiched by the Mondrian pattern by 2 frames at each end, leading to 333 ms presence. The 400-ms-on-400-ms-off pattern was repeated 5 times or until participants reported breakthrough. If breakthrough was reported, the trial ended immediately. If not, another colored word was presented immediately until response. Participants were instructed to name the word (Experiments 1, 3, and 5, 7) or color (Experiments 2, 4, 6, 8) of the target. A 2-alternative-force-choice location task was present at the end of each trial, participants were instructed to report the location of the suppressed prime. While the prime detection served as a subjective report of prime visibility, this location task served as a post-trial objective gauge of prime visibility. The prime was occasionally superimposed on the Mondrians (visible catch) or simply non-existent (blank catch). The primes were blue- or red-colored word forms “BLUE” and “RED” in Experiments 1~8, while the targets were blue- or red-colored word forms “BLUE” and “RED” in Experiments 1~6 and blue- or red-colored word forms “navy” and “scarlet” in Experiments 7~8.

- The interocular suppression phase was repeated five times. What was the design goal of this procedure?

Response 7:

We have shown in the recent study (Hung & Hsieh, under review) that intermittent presentation during the suppression period prolonged the suppression period, allowing us to give a longer total exposure length. This potentially increased the subliminal signal of the stimulus. We have added the explanation in a footnote.

Revision:

P.29. Footnote c.

In a recent study (Hung & Hsieh, under review), we showed that intermittent presentation, coined *discontinuous flash suppression*, during the suppression period delayed breakthrough of the stimulus and hence potentially increased subliminal signal of the stimulus. Thus we applied such interocular suppression paradigm here to achieve longer suppression and exposure duration of the prime.

- In Experiment 5, one subject data look quite abnormal. Without priming, RTs were much slower. Any thought?

Response 8:

We thank the reviewer for pointing this out. The negative dimension in our graph shows *faster* response compared to blank condition. That is, this subject responded faster in both color congruent and incongruent conditions, as compared to blank condition. The subject's raw general reaction time averaged across all conditions was well within 3 standard deviations (1175 ms vs. group (n=20) average 1078 ms with SEM of 55 ms). Also, other objective measures also showed that the subject was decent: visible catch trial accuracy: 100%; blank catch trial accuracy: 96.25%).

Min-Suk Kang

Reviewer #2 (Remarks to the Author):

This manuscript tackles an interesting question: when attention acts on an unseen stimulus does it only affect low-level properties that may have been extracted in early sensory areas, or does it also affect the processing of higher-level semantic properties associated with the stimulus? There is already evidence that attention can act on stimuli that remain unseen, both in terms of enhancing their processing and in terms of selection (the outline of an unseen object can act as the boundary in object-based attention, Norman et al, 2013, Psych Sci.). The experiments presented here are based on the Stroop effect and examine both forward and reverse colour-word Stroop tasks. The effects of both the colour and word-identity of unseen masked primes on reaction times to consciously perceived Stroop stimuli are reported. We already know that unseen colours can act as primes and that their effect can be modulated by attention (Kentridge et al, 2008, Neuropsychologia) and that unseen words can act as primes (see e.g. Rohaut & Naccache, 2018, EJN) so there are good grounds to expect effects of unseen primes on the Stroop task. The most interesting aspect of the design is that it tests whether the 'ink' colour of an unseen prime

influences reaction time is a Stroop task by virtue of its semantics and whether this effect is modulated by attention. Even if it is modulated by attention (and it is), interpretation of the finding is difficult. Should one regard attention as only acting at the stimulus level? For example, is it the processing of the blueness of the ink that is enhanced by attention, but after that, the evocation of blue semantics is only stronger because 'ink blueness' is more strongly represented that it would be should the ink colour have been unattended? Alternatively, should we regard attention itself having enhanced representation of all aspects of the stimulus, both semantic and otherwise?

Response 1:

We agree that the ink color of the invisible word/symbols exerted semantic interference is an interesting result in the current study. As the reviewer pointed out, although we showed that the incongruent color semantics, instead of color percept, slowed down subsequent target response, it is hard to pinpoint at which level exactly did attention modulate the stimulus. The modulation could very well be on the low-level (ink blueness as the reviewer pointed out) or high-level feature (such as semantics as we suggested), or anything in-between. Since we focused on *whether unconscious semantic information can be modulated by attention*, we decided to run additional experiments specifically examine this point. More details of the experimental design, analyses and results will follow in Response 8.

I found it quite hard to understand the precise procedure of these experiments. The legend to figure 1 makes it clear that the location task was 2AFC and that it occurred after the naming task. It also implies that there was a separate opportunity for subjects to report breakthrough from masking. Reports of breakthrough (rather than correct location responses) resulted in the end of a trial. In the methods section on line 579. I presume this is what is referred to as the 'Detection task' in figure 1 (although this term is never used outside the figure itself). I am not clear what the subjects did on the final cycle of priming in a trial (i.e. when they had not reported breakthrough in the detection tasks). Logically they should have also completed a detection task after the final prime presentation. As colour or word naming in the Stroop paradigm is necessarily a reaction time task, I can't see how preceding this with a prime detection task makes any sense. However, on line 581 the authors report that 'If the prime was not detected, the participants reported the word (Exp 1) or colour (Exp 2) of the visible target. This implied that the detection task was completed prior to word of colour naming.

Response 2:

We apologize for not making our experimental procedure clearer. Please see below our revised description and a revised trial sequence figure. We have also revised our manuscript text accordingly.

Revision:

P.29. Line 21-29. P.30. Line 1-4.

In each trial, the participant had three tasks in sequence. All completed via button press. (1) During the suppression period, a detection task is required. The participant was instructed to respond as soon as any part of the word stimulus had been detected. It was stressed that this is the most important task and even seeing a stroke of the word counted as detection. If the prime had been

detected, the trial ended immediately. (2) If the prime was not detected, the participant had to name the color or the word of the visible target word, depending on which experiment they were in. It was stressed that both speed and accuracy were important. (3) At the end of the trial, a two-alternative-force-choice (2AFC) location task asked the participant to judge the location of the suppressed prime. It was made clear that they should try their best to judge even if the prime remained undetected in the suppression period. Please note that this 2AFC location task did not require the participant to hold prime information until the end of the trial but instead served as a reconfirmation of prime invisibility. Thus, the prime invisibility was assessed with both an immediate subjective and a retrospective objective criteria.

Fig. 1. Trial sequence, stimulus, and task. Each trial was self-paced and began with a varied SOA ranging from 0.1~1 s. After which a dynamic flashing Mondrian pattern was presented to the dominant eye while the colored word prime was presented to the non-dominant eye. During 400 ms suppression period, the suppressed word was sandwiched by the Mondrian pattern by 2 frames at each end, leading to 333 ms presence. The 400-ms-on-400-ms-off pattern was repeated 5 times or until participants reported breakthrough. If breakthrough was reported, the trial ended immediately. If not, another colored word was presented immediately until response. Participants were instructed to name the word (Experiments 1, 3, and 5, 7) or color (Experiments 2, 4, 6, 8) of the target. A 2-alternative-force-choice location task was present at the end of each trial, participants were instructed to report the location of the suppressed prime. While the prime detection served as a subjective report of prime visibility, this location task served as a post-trial objective gauge of prime visibility. The prime was occasionally superimposed on the Mondrians (visible catch) or simply non-existent (blank catch). The primes were blue- or red-colored word forms “BLUE” and “RED” in Experiments 1~8, while the targets were blue- or red-colored word forms “BLUE” and “RED” in Experiments 1~6 and blue- or red-colored word forms “navy” and “scarlet” in Experiments 7~8.

It is far from clear what the factors ‘colour consistency’, ‘word consistency’ and ‘target (non)Stroop’ mean? I deduced that ‘colour consistency’, ‘word consistency’ must mean consistency between the colour/word of the prime and the colour/word of the target and that ‘target (non)Stroop’ indicates whether the word and colour of the target word differ (Stroop) or match (non-Stroop). There are many other possibilities though. The manuscript would be far more readable if the meaning of these factors was spelled out.

Response 3:

We thank the reviewer for pointing this out. It is true that there are many other possibilities that we could have categorized the relationship between the prime and target, or within the prime, or within the target. We chose prime-target color and word congruency as well as within target congruency (Stroop/Non-Stroop) to analyze our data due to our hypotheses that responding to different aspects of target modulate how prime-target congruency affects target performance. Now we stick to “congruency” instead of consistency throughout the manuscript for clarity.

Revision:

P.7. Line 8-13.

To examine whether prime-target word and color congruency affects target responses, we put in three different factors into our analysis: *prime-target color congruency*, *prime-target word congruency*, and *target word-color congruency* (that is, whether target is a Stroop word (word-color incongruent) or a non-Stroop word (word-color congruent)). In this article, we refer to prime-target color congruency as color congruency, prime-target word congruency as word congruency, and *target word-color congruency* as target Stroop/Non-Stroop throughout the manuscript.

In Experiment 1 there is an interaction between ‘target (non)Stroop’ and ‘colour consistency’ but not with ‘word consistency’. That suggests that the prime can act to enhance the effect of the Stroop distractor dimension (colour in a naming task), but it doesn’t affect the task dimension (word naming in this case). If the colour of the prime and the colour of the target both don’t match the target word in a word naming task RTs are slowest. If the colour of the prime and the target match and the target is non-Stroop – so the colour of the prime and target match the target word, RTs are fastest. I am not sure why this is especially surprising if word reading of the target is so effortless and automatic as to be unaffected by the prime (such reading automaticity is meant to be the basis of the Stroop effect).

In Experiment 2, the classic Stroop colour naming task, this is a Stroop effect, but there are no effects of colour or word consistency between prime and target. Colour naming is effortful and far from automatic. We might expect the name of the target word to be processed automatically (that is what generates the Stroop effect) and this to be so fast that it cannot be made faster by a matching prime or slowed by a distracting one. Once colour naming becomes more practiced the effect of the distractor dimension (word name) becomes stronger (reaching statistical significance). So here when the primary task is effortful, we find no effect of primes, but once it becomes easier the distractor dimension primes exert an effect. Exactly as they do in the word-reading task.

Something has gone wrong around lines 215 to 233. Some blocks of text appear twice. There is a non-sequitur at line 223 (I don’t think ‘colour naming...’ really follows on from its preceding text).

Response 4:

We apologize for the repetition, which has been removed in the revision.

In the discussion of this section the authors claim that “Experiment 1 showed both word-induced and colour-induced semantic interference between a suppressed distractor and a visible target” (line 251). I don’t see the evidence for an effect of word consistency on the (reverse) Stroop effect in this experiment. There is no interaction between ‘colour-consistency’ and ‘target (non) Stroop’. There is an interaction between word and colour consistency of the primes, but that effect occurs independent of the Stroop effect. The post-hoc comparisons reported show that it is colour congruency that leads to very slow RTs for Stroop targets and very fast RTs for non-Stroop targets. Word-consistency is playing no role that I can see.

Response 5:

We believe that this confusion came from our writing of the manuscript. And such confusion interfered with how the reviewer interpreted our results.

Similar to our reply to an earlier comment, word and color consistency (now congruency) refer to the consistency *between* prime and target, and Stroop/non-Stroop refers to within target word-color consistency. Therefore, the interaction between word and color consistency between prime and target showed that both word and color incongruency contributed to the slowing of the target response.

Experiments 3 and 4 repeat the preceding Stroop and reverse-Stroop experiments but use colourless (grey I suppose) primes in order to distinguish between the effects of prime colour and semantics. In addition, half of the trials contain no prime and so can be used as a baseline (although the mere presence of a prime might have an effect of RTs via alerting, so perhaps a better baseline control would use irrelevant words or non-word primes). In the reverse Stroop experiment (Expt 3) there is a Stroop effect and also an effect of prime congruency. I don’t know why these analyses are not presented as an ANOVA with factors ‘prime congruency’ and ‘target (non) Stroop’, as before. An effect of prime congruence shows that matches between the prime word and target word speed RTs regardless of the colour in which the target word is written. Word-naming is the task dimension of this reverse Stroop task and so we might be surprised that, in contrast to the previous experiments, we find an effect in the task-dimension, but not in the distractor-dimension. However, without conducting an ANOVA we cannot know whether this prime congruency had different effects on Stroop and non-Stroop targets. What has been demonstrated is an effect of word-identity priming – something we shouldn’t be too surprised to find. Experiment 4, a colour naming task, again examines early and late trials separately. In this experiment an effect of prime-target congruence is found in the later trials (here, the distractor-dimension of the Stroop task), but no attempt is made to test whether there was a differential effect on Stroop and non-Stroop trials.

Response 6:

We thank the reviewer to point this out. We did not apply the same ANOVA analysis because the numbers we are handling here are the de-baselined RT percentage *differences*. In Experiments 3~6, we added in blank trials to serve as the baseline. As the reviewer pointed out, the mere presence of a prime may have an alerting effect, so we figured that to examine if the effect was different between the incongruent and congruent prime, we could compare the trials in the same condition *with* versus *without* a suppressed prime. That is, testing if the presentation of a congruent or

incongruent prime made the same trials, without the prime, faster or slower. So intrinsically the Stroop effect could be deducted from the baseline. In this way, we could specifically focus on the congruency/incongruency effect. We have made this point clearer in our manuscript.

However, as the reviewer pointed out the prime may have differential effect on Stroop and non-Stroop trials on top of the RT normalization, thus we also performed the 2-way repeated measure ANOVA with the raw reaction time. Target Stroop/Non-Stroop did not interact with the congruency effect. The results have been added to supplementary information and copied below.

Revision:

Supplementary information

1. Two-way ANOVA analyses on the raw reaction time in Experiments 3~6.

Experiment 3

To specifically focus on the congruency/incongruency, we utilized our prime-blank trials as the baseline trials and calculated the reaction time percentage differences between these baseline trials and the trials with congruent and incongruent primes. We believe that comparing identical targets with and without the prime could reveal a more direct effect from the suppressed prime. Therefore, the following results will appear in reaction time percentage changes from the baseline trials.

To examine if the congruency effect interacts with target reverse Stroop/non-Stroop, we also performed a two-way (word consistency; target (non)Stroop) repeated measures analysis of variance on the target word raw reaction time. We found significant main effects of target reverse Stroop/non-Stroop ($F(1, 19) = 9.25, p = .007, \eta_p^2 = .33.$) and prime-target word congruency ($F(1, 19) = 32.99, p = .00001, \eta_p^2 = .27.$). The interaction between the two effects was not significant with $F(1, 19) = 0.09, p = .77, \eta_p^2 = .00.$

Experiment 4

Similarly, we also performed a two-way (word consistency; target (non)Stroop) repeated measures analysis of variance on the target word raw reaction time in all trials, 1st quarter trials, and 4th quarter trials. In all trials, we found significant main effect of target Stroop effect ($F(1, 19) = 19.83, p = .0003, \eta_p^2 = .51.$) but not prime-target word congruency ($F(1, 19) = 1.59, p = .22, \eta_p^2 = .08.$). The interaction between the two effects was not significant with $F(1, 19) = 1.78, p = .20, \eta_p^2 = .02.$ Same analysis performed on 1st quarter trials showed similar results: significant main effect of target Stroop effect ($F(1, 19) = 13.62, p = .002, \eta_p^2 = .42.$) with no effect on prime-target word congruency ($F(1, 19) = 0.10, p = .75, \eta_p^2 = .01.$). The interaction between the two was not significant with $F(1, 19) = 0.07, p = .80, \eta_p^2 = .00.$ On the contrary, in the 4th quarter trials, there were significant main effects of both target Stroop/non-Stroop ($F(1, 19) = 7.01, p = .02, \eta_p^2 = .27.$)

and prime-target word congruency ($F(1, 19) = 4.87, p = .04, \eta_p^2 = .20$.) with no interaction between the two ($F(1, 19) = 0.001, p = .98, \eta_p^2 = .00$).

Experiments 5 and 6 have modifications complementing those of experiments 3 and 4. The primes now have colour content, but no semantic content (the 'words' and always XXXX). The analysis presented are now based on the 'colour reading' and 'word reading' accounts presented at line 316 and following. It is not clear at line 354 whether the results being presented are from experiment 5 or from some combination of experiments 5 and 6 although all becomes clear a little later. The analyses are now made in terms of the congruence between the prime colour and target colour or between the prime colour and the target word. Evidence is found for an effect of congruence between prime colour and target word but there is no evidence for an effect of congruence between prime colour and target colour. The task is word reading so the distractor dimension is colour. We see an effect of the distractor dimension but not the task dimension. Again, as no ANOVA is presented, we do not know if this effect modulates the Stroop effect.

In experiment 6 (colour naming) there were no effects of congruency between prime colour and target colour of word (even when trials were split into early and late blocks). The distractor dimension in a colour-naming task is word identify and, of course, that is not a variable in this experiment. Again, the analysis does not permit interactions between prime consistency and target (non) Stroop to be analysed.

Response 7:

Similar to additional analyses on raw reaction time in Experiments 3 and 4, we also performed the 2-way ANOVA in Experiments 5 and 6 to see if there is interaction between target-prime color consistency and target (non)Stroop. Target Stroop/Non-Stroop did not interact with the congruency effect. Please see the detailed results below.

Revision:

Supplementary information

1. Two-way ANOVA analyses on the raw reaction time in Experiments 3~6.

Experiment 5

To look into the interaction between prime-target congruency and target (non)Stroop, we also performed a two-way (color consistency; target (non)Stroop) repeated measures analysis of variance on the target word raw reaction time in Experiment 5. Similar to what we found with reaction percentage changes normalized with baseline blank trials, grouping trials according to the color-perceiving account showed only the main effect on the target Stroop effect ($F(1, 19) = 23.03, p = .0001, \eta_p^2 = .55$, with neither main effect on the color congruency ($F(1, 19) = 0.07, p = .80, \eta_p^2 = .00$, nor the interaction between the two ($F(1, 19) = 0.46, p = .51, \eta_p^2 = .01$). similarly, grouping trials based on the color-reading account showed only the main effect on the target Stroop effect ($F(1, 19) = 23.03, p = .0001, \eta_p^2 = .55$, with neither main effect on the color congruency

($F(1, 19) = 0.46, p = .51, \eta_p^2 = .02$, nor the interaction between the two ($F(1, 19) = 0.07, p = .80, \eta_p^2 = .00$).

Experiment 6

In Experiment 6, we also performed a two-way (color consistency; target (non)Stroop) repeated measures analysis of variance on the target word raw reaction time in all trials, 1st quarter trials, and 4th quarter trials. In all trials, we found significant main effect of target Stroop effect ($F(1, 19) = 37.37, p = .00, \eta_p^2 = .66$.) but not prime-target color congruency ($F(1, 19) = 0.93, p = .35, \eta_p^2 = .05$.). The interaction between the two effects was not significant with $F(1, 19) = 0.73, p = .40, \eta_p^2 = .01$. Same analysis performed on 1st quarter trials showed similar results: significant main effect of target Stroop effect ($F(1, 19) = 30.96, p = .00, \eta_p^2 = .62$.) with no effect on prime-target color congruency ($F(1, 19) = 0.01, p = .91, \eta_p^2 = .00$.). The interaction between the two was not significant with $F(1, 19) = 1.15, p = .30, \eta_p^2 = .02$. Similarly, in the 4th quarter trials, there was significant main effect on target Stroop/non-Stroop ($F(1, 19) = 18.74, p = .00, \eta_p^2 = .50$.) but not prime-target color congruency ($F(1, 19) = 0.06, p = .81, \eta_p^2 = .00$.) with no interaction between the two ($F(1, 19) = 0.06, p = .81, \eta_p^2 = .00$).

The aim of all of these experiments has been to determine whether attention, as manipulated by the demands of the Stroop task, can affect 'high-level' visual information. The first two experiments certainly show that the primary task determines whether colour or word congruency between prime and target affects performance. In both cases only the distractor dimension of the prime affects the conscious Stroop effect. In the colour-naming (classic Stroop) task this effect only becomes evident once the task is well practiced. It is, of course, only through the meaning of the prime colour in the reverse-Stroop experiment (Expt 1) that an effect can be exerted. This seems good evidence that when an unseen stimulus is selected by attention the effect of its semantic properties on subsequent conscious decisions can be enhanced. I am not, however, sure whether, on this basis, one can claim that attention is operating on high-level unconscious visual information.

Point well received. It could be attention enhancement of low-level stimulus feature which then enhances a semantic effect.

It is much harder to make the claim that the effects of attention on word-congruence act at a semantic level. Congruence here implies identity between letters in prime and target and so the priming effect might operate at a level prior to reading (showing unconscious modulation of Stroop effects using colour-related words like 'sky', 'grass', or 'fire' would be much stronger evidence). I am not sure that the subsequent experiments added that much to the findings and I am extremely surprised that they were not analysed with ANOVAs so as to allow priming effects on the modulation of visible Stroop effects to be analysed. The authors' 'load theory' explanation of the development of priming over trials in colour naming tasks makes a great deal of sense. Unlike word reading, the colour naming task is demanding, and it is unsurprising that it is only affected by primes once it has become less practiced and requires less effort.

One concern I have is whether attention really is operating on high-level unconscious visual information in these experiments. Both revolve around the question of what it means for an item to be attended? Should an item be considered attended simply if attention enhances its processing or should only be thought of as attended if it is selected by attention (see e.g. Mole, 2008, *Journal of Consciousness Studies*, but see also Norman et al, 2013, *Psych Science*). In spatial attention we may attend to a region of space and mental resources may be allocated to items within that region of space, even though the items themselves were not selected. Are the items attended or is it only the space they occupy that is attended? Here, task demands are driving attention to colour or to words, and the processing of specific properties of the primes are being enhanced, but, in analogy with Mole's argument, one might claim that it is not blueness that is being selected when attention is directed towards colour in general. When processing of the unseen blue colour is enhanced it is enhanced by virtue of being a colour, so selection is not based on its semantics. Cueing conscious feature-based attention to blueness and then showing that semantic properties of an unseen blue stimulus (e.g. showing that as the unseen blue item was star shaped it primed responses to the visible word 'rocket' would surely qualify).

Response 8:

We thank the reviewer to point out this important question. Indeed, is it difficult to pinpoint if word-induced incongruency effect from an invisible prime was at the semantic level. Another equally important issue is what is the nature of attention modulation in our study? Is it based on the task set alignment which triggered feature-selective attention to word or color aspect of the unconscious prime? Or it is based on the available general attentional resources competing between an unconscious prime and a conscious target? Each of the two questions cannot be simply answered by the first six experiments in that (1) as the reviewer pointed out, task load and task set manipulation were intertwined in these experiments. Word-naming, throughout these experiments, was of lower task load and oriented participants to the semantic feature of the target. On the other hand, color-naming was of higher task load and prompted attention to be directed to color feature of the target. As we are focusing on the semantic interference from the unconscious prime, word-naming will be considered with aligned task set while color-naming will be with unaligned task set. (2) Since the primes and targets had identical word forms, the attention modulation could operate on the orthographic or the semantic relationship between the prime and target, or both.

In order to conclusively show that semantic congruency can be modulated by how participants distribute their attention in the task, we performed two more experiments. In Experiments 7 & 8, we chose two new lowercase words as the visible targets "scarlet" and "navy" in blue or red (scarlet, navy) while the suppressed primes remains colorless "BLUE" or "RED" so that these words still exhibited a semantic but not orthographic (dis)similarity **with the prime**. Importantly, in our pilot data (n = 2), word-naming was still accompanied by an interference from the target color (reverse Stroop effect), while the typical Stroop effect from word semantic interference during color-naming disappeared. This opposite pattern of the strength of Stroop/reverse Stroop effects was later confirmed with twenty participants in each experiment. With strong reverse Stroop effect and no Stroop effect on the targets, our new experiments disentangled the task load/task set relationship. Word-naming is now characterized as high load (due to the reverse Stroop effect) yet with aligned task set, while color-naming is now low load (due to the disappearance of Stroop effect) with unaligned task set. Two separate pairs of conflicting hypotheses can be articulated on (1) the modulated feature: orthography or semantics and (2) the modulating source: task load or task set. In (1), if pure semantics from an unconscious prime could

exert an effect, we expected to see unconscious interference on the target responses. Otherwise if the interference effect was mainly driven by orthographic similarities, the effect will disappear. In (2) if task load played a key role in modulating the interference, the interference would appear in the new low load experiment (Exp 8, color-naming) but not the new high load experiment (Exp 7, word-naming, or only in the later practiced trials). Otherwise if task set alignment played a key role, the interference would appear in the word naming experiment (Exp 7), but not the color naming experiment (Exp 8).

Our results showed that interference between prime and target still persisted, suggesting that pure semantics between prime-target could exert an effect. Moreover, this semantic interference only occurred in the low load (Exp 8) but not high load experiment (Exp 7), suggesting that task load, but not task set alignment, was the determining modulator.

Firstly, we observed opposite patterns of Stroop/reverse-Stroop effects. In Experiment 7 when the participants were instructed to name the word, a significant reverse Stroop effect arose ($t(19) = 3.15$, $p = 0.005$, Cohen's $d_{av} = 0.32$), while in Experiment 8, during color-naming, the typically stronger Stroop effect completely disappeared ($t(19) = 0.85$, $p = 0.41$, Cohen's $d_{av} = 0.09$). This showed that word-naming experiment was of high load while color-naming was of low load, which was the opposite of typical Stroop/reverse Stroop effects.

In Experiment 7, the semantic congruency effect was neither significant in all trials ($t(19) = 0.82$, $p = .42$, Cohen's $d_{av} = 0.19$) nor the later trials ($t(19) = 1.21$, $p = 0.24$, Cohen's $d_{av} = 0.33$). On the other hand, such effect was significant in Experiment 8 in all trials with an interaction with prime location. That is, only when the invisible prime and visible target were in the same side of fixation point (locations were still jittered), an invisible prime (e.g. BLUE) prompted a slower response to a subsequent visible semantically incongruent target (e.g. Scarlet), as compared to a congruent one (e.g. Navy).

How do these two new experiments shed light on our interpretation of the previous findings? First of all, we show that pure conceptual/semantic incongruency between an unconscious prime and a conscious target exerted a slowing interference effect on target response. Furthermore, this interference was modulated by *task load*. When task load is low (color-naming, no Stroop effect), the interference arises. On the contrary, when task load is high (word-naming, reverse Stroop effect), the unconscious interference disappears. We could thus pinpoint that task load, but not task set alignment, played a key role in modulating unconscious semantic interference.

We have added in the new experiments, detailed analyses, and discussions into the manuscript.

Revision:

Introduction

P.5. Line 21-29.

..... Meanwhile, responding to different aspects of a target word could selectively activate feature-specific attention, which could be further utilized to select the subliminal relevant feature. That is, responding to the semantic feature of a supraliminal word could selectively deploy semantic-specific attention to the subliminal semantic feature. Two hypotheses can thus be raised

and examined. (1) An unconscious effect from high-level visual content requires *general attentional resources*. That is, an unconscious effect will occur only when the task load is low and additional attentional resources could be distributed to the unconscious stimulus. (2) An unconscious effect from high-level visual content requires *feature-specific attention*. That is, an unconscious effect will occur only when the task set is tuned to the critical subliminal feature.

Results

P. 17-20 and table on P.21

Attention modulation on unconscious semantic interference: task load or task set? semantic congruency or orthographic similarity?

In Experiments 1~4, we found that when the subliminal prime and supraliminal target were semantically incongruent (e.g. BLUE vs. RED), the incongruency prompted slower target responses. This effect was further shown to be modulated by task load. When the task load was low (Experiments 1 & 3, word naming), the effect appeared in all trials. When the task load was high (Experiments 2 & 4, color naming), the effect appeared only in the later trials when there was a practice effect on the task. This effect was interpreted as a word-induced subliminal semantic interference. However, as the subliminal primes and targets had identical visual forms (BLUE/RED), one may argue that the effect could simply emerge from low-level visual feature adaptation. Although in all experiments, the prime and target were of different font sizes and presented on jittered locations. We first proceeded to examine if prime-target co-localization (on the same or different side of the fixation point) interacted with the word congruency effect with the reasoning that if stronger effect arose when the prime and target were spatially closer, low-level visual adaptation may contribute significantly to our word incongruency effect. We thus ran an additional two-way (word congruency; prime-target location consistency) repeated measures analysis of variance on the data of Experiments 3 & 4. The results from both experiments showed no interaction between location and semantic congruency (Experiment 3, all trials, $F(1, 19) = 0.41$, $p = 0.53$, $\eta_p^2 = .004$; Experiment 4, 4th quarter trials, $F(1, 19) = 0.48$, $p = 0.50$, $\eta_p^2 = .01$). Such results suggested that our findings that prime-target word incongruency slowed down target responses were not merely driven by low-level adaptation of visual stimulus shapes.

However, since our unconscious primes and conscious targets shared not only semantic but also orthographic similarities, it is almost impossible to tease apart if such effect was due to word form (orthographic) incongruency or semantic/conceptual incongruency, or both. To directly examine whether task-induced attention could modulate pure semantic interference, we adopted the design in Experiments 3 and 4 with one adjustment: replacing the target words to “scarlet” and “navy.” By utilizing these two target words, the prime and target now contained no orthographic similarities while still exhibited semantic/conceptual (in)congruencies. It has been shown that once the color association between colors and words is weaker (e.g. naming the color blue on the word *red* vs. *and*), the Stroop interference from color-naming weakens²⁰. The authors also proposed that word frequency could play a key role as low word frequency is typically linked to weaker association with colors. This early observation led to the prediction in our current study that color-naming on the words *navy* and *scarlet* would accompany weaker word semantic interference (i.e. weaker Stroop effect) as demonstrated in the original study. Indeed, in our pilot data ($n = 2$), this choice of lower-frequency words as targets reversed the relative task load for word naming and color naming: Word-naming was still accompanied by an interference from the color of the target

(reverse Stroop effect), while the typical Stroop effect from color-naming disappeared. This opposite pattern of the strength of Stroop/reverse Stroop effects was later confirmed with twenty participants in each experiment. Such pattern teased apart the tight relationship between task load and task set alignment in our previous experiments. That is, the stronger unconscious prime semantic incongruency effect in word-naming experiments can be attributed to either lower task load or task set alignment (attending to semantic feature). While weaker semantic incongruency effect in color-naming can be due to high task load or task set unalignment (attending to non-semantic feature). Thus, it is again almost impossible to pinpoint whether task-induced attentional load or feature-specific attention modulated the unconscious effect. Nonetheless, with the current design, we have created a high-load x task-effect aligned feature-attending (word naming) and low-load x task-effect unaligned feature-attending (color-naming). The new experiments allowed us to examine two separate pairs of conflicting hypotheses on (1) the modulated feature: orthography or semantics and (2) the modulating source: task load or task set. In (1), if pure semantics from an unconscious prime could exert an effect, we expected to see unconscious interference on the target responses. Otherwise if the interference effect was mainly driven by orthographic similarities, the effect would disappear. In (2), if task load played a key role in modulating the interference, the interference would appear in the new low load experiment (Experiment 8, color-naming) but not the new high load experiment (Experiment 7, word-naming, or only in the later practiced trials). Otherwise if task set alignment played a key role, the interference would appear in the word-naming but not the color-naming experiment.

Firstly, we again obtained similar data on successful interocular suppression in both experiments (Experiment 7: location task accuracy 49.81% (1.15%), paired $t(19) = -0.16, p = 0.87$; Experiment 8: 50.2 % (1.33 %), paired $t(19) = 0.15, p = 0.88$). Moreover, the near-ceiling performance was found on the blank and visible trials with 97.78% (0.51%) and 98.12% (1.29%) accuracies in Experiment 7 and 97.81% (0.54%) and 99.69% (0.31%) accuracies in Experiment 8. The critical difference was that the reverse Stroop effect was evident in Experiment 7 (Reaction time comparison of Stroop and non-Stroop trials: $t(19) = 3.15, p = 0.005$, Cohen's $d_{av} = 0.32$) while the Stroop effect was not in Experiment 8 (Reaction time comparison of Stroop and non-Stroop trials: $t(19) = 0.85, p = 0.41$, Cohen's $d_{av} = 0.09$).

In Experiment 7, the RTs in each condition were first normalized against blank trials. A direct paired comparison between semantically congruent and incongruent trials showed null effect ($t(19) = 0.82, p = .42$, Cohen's $d_{av} = 0.19$, Fig. 6A). On average, compared to the blank trials, semantic congruent trials were 2.1 % slower while incongruent ones 0.9 % were slower. We also examined whether prime-target location (same-different) had interacted with the semantic congruency effect. A two-way (semantic congruency; prime-target location consistency) repeated measures analysis of variance was performed on the normalized target word reaction time. Neither the main effect of semantic congruency, $F(1, 19) = 0.09, p = 0.77, \eta_p^2 = .005$, nor the main effect of location congruency was found, $F(1, 19) = 1.32, p = 0.26, \eta_p^2 = .07$. There was no interaction between the two, $F(1, 19) = 0.21, p = 0.65, \eta_p^2 = .01$. Similar results were found in the 4th quarter trials (direct comparison: $t(19) = 1.21, p = .24$, Cohen's $d_{av} = 0.33$; semantic congruency x location consistency 2-way ANOVA: main effect of semantic congruency $F(1, 19) = 0.07, p = 0.80, \eta_p^2 = .004$; main effect of location: $F(1, 19) = 0.11, p = 0.75, \eta_p^2 = .005$; interaction: $F(1, 19) = 0.49, p = 0.49, \eta_p^2 = .03$.)

In Experiment 8. A direct paired comparison on the normalized RT between semantically congruent and incongruent trials showed null effect ($t(19) = -1.12, p = .28$, Cohen's $d_{av} = 0.24$). On average, compared to the blank trials, semantic congruent trials were 1.32 % slower while incongruent ones 2.7 % were slower. However, a two-way (semantic consistency; prime-target location consistency) repeated measures analysis of variance showed an interaction between the prime-target location and semantic congruency with $F(1, 19) = 8.13, p = 0.01, \eta_p^2 = 0.30$. The main effects were not significant (semantic congruency: $F(1, 19) = 1.1, p = 0.29, \eta_p^2 = 0.06$; location congruency: $F(1, 19) = 0.05, p = 0.83, \eta_p^2 = 0.003$). A planned post hoc comparison between semantic congruent and incongruent trials when the prime-target were co-localized showed a significant effect with $t(19) = -2.45, p = .02$, Cohen's $d_{av} = 0.63$, Fig. 6B).

Fig. 6. Experiments 7 and 8 results. Y-axis denotes reaction time, and X-axis denotes two critical conditions: semantically congruent (SEM-CON) and semantically incongruent (SEM-INCON). Black dots and lines denote longer RT in the semantically incongruent condition while gray dots and lines denote longer RT in the semantically congruent condition. Each pair of dots lined between the two conditions represents one participant. Upward lines indicate longer RT increase in the incongruent condition while downward lines indicated RT decrease in the incongruent condition. The bars denote group mean with the error bars indicating standard error of the mean (SEM). Asterisk denotes significance. Please note that the data from Experiment 8 was from prime-target co-localized trials.

These results showed that, without orthographic similarity, the semantic incongruency between the prime and target did pose a slowing effect on the target responses if that they were co-localized. The observation that the semantic effect further interacted with prime-target co-localization indicated the need of spatial attention toward the primed location for such effect to occur. This semantic effect was also modulated by the task load. In Experiment 8 (color-naming), when the task load was low as indicated by the lack of the Stroop effect, the semantic incongruency slowed down target response. On the contrary, in Experiment 7 (word-naming), when the task load was high as indicated by the reverse Stroop effect, such effect disappeared. As task alignment and task load were separated in these two experiments (i.e. Experiment 7: Semantic task but high load, Experiment 8: Non-semantic task but low-load), distinct from all previous experiments. This particular finding stressed that the level of task load, but not task alignment, modulated an unconscious semantic effect in our paradigm. Table 1 summarizes the stimulus type, task, and effect in all experiments.

Experiment	Prime	Target	Task	Load	Interference
1	BLUE BLUE RED RED		Word-naming	Low	Yes
2	BLUE BLUE RED RED		Color-naming	High	Yes after practice
3	BLUE RED	BLUE BLUE	Word-naming	Low	Yes
4	BLUE RED	RED RED	Color-naming	High	Yes after practice
5	XXXX XXXX		Word-naming	Low	Yes
6	XXXX XXXX		Color-naming	High	No
7	BLUE RED	navy navy	Word-naming	High	No
8	BLUE RED	scarlet scarlet	Color-naming	Low	Yes

Table 1. The table summarizes all eight experiments where we manipulated the color, word, and semantic relationship between the subliminal prime and supraliminal target. Task (word-naming vs. color-naming) and task load (high vs. low, according to the strength of Stroop/Reverse Stroop effect) in each experiment are presented.

Discussion

P.22 Line 8-32. P.23. Line 1-7.

Two important questions were left answered after our first six experiments. Firstly, although the general predictions of the load theory are compatible with our findings, task-induced attention may *selectively* trigger different cognitive sets under different tasks, which in turn gates how an unconscious prime interfered with subsequent target performance. For example, in the word-naming experiments (Experiments 1, 3, & 5), participants' attention was selectively oriented to the word semantics. Such attention tuning would have been applied further to the suppressed prime and allowed semantic interferences to occur. On the contrary, in the color-naming experiments, attention was deployed to the word color, which would have interfered the semantic processing of the target (i.e. the Stroop effect) as well as hindered the semantic interferences from the suppressed prime. Since both our interfering effects were semantic in nature, word- and color-induced semantic congruency effects were gated by this cognitive-set-induced-tuning to different levels: Under word-naming task, both exerted an effect; while under color-naming task, only the more automatic word-induced incongruency effect re-emerged after the participants became more fluent in the task (i.e. 4th quarter trials in Experiments 2 & 4). Secondly, one could argue that since identical word-forms were used as primes and targets, what appeared to be a semantic incongruency effect could be driven merely by orthographic dissimilarity. Although we had jittered the prime-target locations and font sizes, and our further analyses focusing on the interaction between the semantic effect and location yielded null effect, which excluded the possibility of low-level adaptation.

Thus, (1) to ensure that sheer semantic effect could be modulated and (2) to tease apart cognitive-set-induced-tuning and task load modulation, in Experiments 7 and 8, we went one step further to remove the orthographic similarity of the prime and target while maintaining the semantic relationship by introducing a different set of targets: *navy* & *scarlet*. Importantly, the Stroop but not the reverse Stroop effect disappeared when participants were instructed to name the color and word, respectively. Such manipulation thus enabled a misalignment between cognitive-set-induced-tuning attention and task load, allowing us to pinpoint the determining factor in

modulating the unconscious effect. As a result, we showed the unconscious semantic incongruency effect when the task load was low yet non-semantic (Experiment 8, color-naming) but not when the task load was high but semantic (Experiment 7, word-naming), suggesting that general availability of attentional resources, but not cognitive-set-induced-tuning attention, played a key role in modulating the unconscious semantic interference in our study.

Even if we reject this argument out of hand, we might still worry whether attention really is operating on high-level unconscious visual information in these experiments. The blue coloured word (or XXXX non-word) is attended, but can we claim that it is attention that is operating on its semantic properties? We know that the semantics of unseen items are processed (and indeed that semantic relatedness between unseen primes and visible target affects performance – van der Bussche et al, 2009, Experimental Psych) so is attention specifically responsible for semantic access here?

Overall I felt experiments 1 and 2 were interesting but I was not sure that the rest of the experiments were really necessary, especially when reported and analysed in a form that does not show how the primes are affecting the consciously attended and unattended dimensions of the target to which responses are made. It is clear that conscious task demands determine which properties of unseen primes affect behaviour. For colour primes it is also clear that these behavioural effects are mediated by the semantics of the attended unseen prime. I think more care has to be taken in arguing that attention is specifically acting on the unseen prime's semantic properties though.

BOB KENTRIDGE

Reviewer #3 (Remarks to the Author):

This very interesting study investigated the effect of task-induced attention (or maybe more accurately described as cognitive set) on the potential interference from invisible information. The authors devised an innovative double Stroop paradigm, with the first presentation rendered invisible using interocular suppression. Results convincingly showed a semantic level interference effect from the first stimulus that was modulated by the task as well as the attentional load on the second stimulus. The experiments are solid and the results are interesting. I would like the authors to consider and clarify the following issues (#3 is the more critical):

1. conceptually, the current study is quite similar with Kiefer & Martens (2010), even though the two studies were conducted with different methodologies. The authors cited this paper, but I believe it deserves more discussion, especially in terms the novel insights from the current study beyond the Kiefer & Martens study.

Response 1:

We agree that our findings from the first six experiments led to similar conclusions as Kiefer & Martens (2010). Instead of attributing to task load, one could argue that *cognitive/task set* could modulate the semantic interference from an unconscious prime in our experiments. This is because in our original six experiments, due to the nature of the Stroop task, cognitive set alignment (semantic effect x word-naming) was always associated with low-load (reverse Stroop effect)

while cognitive set unalignment (semantic effect x color-naming) was associated with high-load (Stroop effect). However, our additional experiments addressed this issue and showed that task load, instead of task set, modulated the semantic interference to the response of target Stroop/reverse Stroop task. More details on the experimental design, analyses, and results are shown in Response 2. We have also added in more discussion on the difference of our study.

2. while the authors interpreted their results as effect from "task-induced attention", it seems a better description is cognitive set. At least it should be more clearly described as "feature" attention.

3. there is a potential confusion about where was the gating effect of attention/cognitive set applied. The title suggests that the authors believe the gating was on "unconscious semantic interference", but in the abstract and elsewhere, they seemed to suggest that their results "demonstrate the need of attention in extracting unconscious information". These two interpretations are fundamentally different. It is possible that EXTRACTING unconscious information is less dependent on attention/cognitive set, but the interference effect from the extracted information on the subsequent task is contingent on feature attention/set.

Response 2:

In comment 2, we believe that the reviewer pointed out another potential explanation of our data. Task-induced attention in our previous manuscript referred specifically to task-induced attentional load. That is, color-naming task induced a higher task load than word-naming task, which in turn gated how an unconscious prime interfered with subsequent target performance. Another possible explanation to our results was that color-naming task and word-naming task triggered completely different cognitive sets and thus gated unconscious information differently. These two possibilities cannot be teased apart by the first six experiments in that as the reviewer pointed out, task load and task set manipulation were intertwined in these experiments. Word-naming, throughout these experiments, was of lower task load and oriented participants to the semantic feature of the target. On the other hand, color-naming was of higher task load and prompted attention to be directed to color feature of the target. In comment 3, we also acknowledged that our first six experiments did not allow us to pinpoint what particular unconscious information was modulated by attention. Since the primes and targets had identical word forms, the attention modulation could operate on the orthographic or the semantic relationship between the prime and target, or both.

In order to conclusively show that semantic congruency can be modulated by how participants distribute their attention in the task, we performed two more experiments. In Experiments 7 & 8, we chose two new lowercase words as the visible targets "scarlet" and "navy" in blue or red (scarlet, navy) while the suppressed primes remains colorless "BLUE" or "RED" so that these words still exhibited a semantic but not orthographic (dis)similarity **with the prime**. Importantly, in our pilot data (n = 2), word-naming was still accompanied by an interference from the target color (reverse Stroop effect), while the typical Stroop effect from word semantic interference during color-naming disappeared. This opposite pattern of the strength of Stroop/reverse Stroop effects was later confirmed with twenty participants in each experiment. With strong reverse Stroop effect and no Stroop effect on the targets, our new experiments disentangled the task load/task set relationship. Word-naming is now characterized as high load (due to the reverse Stroop effect) yet with aligned task set, while color-naming is now low load (due to the disappearance of Stroop effect) with unaligned task set. Two separate pairs of conflicting

hypotheses can be articulated on (1) the modulated feature: orthography or semantics and (2) the modulating source: task load or task set. In (1), if pure semantics from an unconscious prime could exert an effect, we expected to see unconscious interference on the target responses. Otherwise if the interference effect was mainly driven by orthographic similarities, the effect will disappear. In (2) if task load played a key role in modulating the interference, the interference would appear in the new low load experiment (Exp 8, color-naming) but not the new high load experiment (Exp 7, word-naming, or only in the later practiced trials). Otherwise if task set alignment played a key role, the interference would appear in the word naming experiment (Exp 7), but not the color naming experiment (Exp 8).

Our results showed that interference between prime and target still persisted, suggesting that pure semantics between prime-target could exert an effect. Moreover, this semantic interference only occurred in the low load (Exp 8) but not high load experiment (Exp 7), suggesting that task load, but not task set alignment, was the determining modulator.

Firstly, we observed opposite patterns of Stroop/reverse-Stroop effects. In Experiment 7 when the participants were instructed to name the word, a significant reverse Stroop effect arose ($t(19) = 3.15$, $p = 0.005$, Cohen's $d_{av} = 0.32$), while in Experiment 8, during color-naming, the typically stronger Stroop effect completely disappeared ($t(19) = 0.85$, $p = 0.41$, Cohen's $d_{av} = 0.09$). This showed that word-naming experiment was of high load while color-naming was of low load, which was the opposite of typical Stroop/reverse Stroop effects.

In Experiment 7, the semantic congruency effect was neither significant in all trials ($t(19) = 0.82$, $p = .42$, Cohen's $d_{av} = 0.19$) nor the later trials ($t(19) = 1.21$, $p = 0.24$, Cohen's $d_{av} = 0.33$). On the other hand, such effect was significant in Experiment 8 in all trials with an interaction with prime location. That is, only when the invisible prime and visible target were in the same side of fixation point (locations were still jittered), an invisible prime (e.g. BLUE) prompted a slower response to a subsequent visible semantically incongruent target (e.g. Scarlet), as compared to a congruent one (e.g. Navy).

How do these two new experiments shed light on our interpretation of the previous findings? First of all, we show that pure conceptual/semantic incongruency between an unconscious prime and a conscious target exerted a slowing interference effect on target response. Furthermore, this interference was modulated by *task load*. When task load is low (color-naming, no Stroop effect), the interference arises. On the contrary, when task load is high (word-naming, reverse Stroop effect), the unconscious interference disappears. We could thus pinpoint that task load, but not task set alignment, played a key role in modulating unconscious semantic interference.

We have added in the new experiments, detailed analyses, and discussions into the manuscript.

Revision:

Introduction

P.5. Line 21-29.

..... Meanwhile, responding to different aspects of a target word could selectively activate feature-specific attention, which could be further utilized to select the subliminal relevant feature.

That is, responding to the semantic feature of a supraliminal word could selectively deploy semantic-specific attention to the subliminal semantic feature. Two hypotheses can thus be raised and examined. (1) An unconscious effect from high-level visual content requires *general attentional resources*. That is, an unconscious effect will occur only when the task load is low and additional attentional resources could be distributed to the unconscious stimulus. (2) An unconscious effect from high-level visual content requires *feature-specific attention*. That is, an unconscious effect will occur only when the task set is tuned to the critical subliminal feature.

Results

P. 17-20 and table on P.21

Attention modulation on unconscious semantic interference: task load or task set? semantic congruency or orthographic similarity?

In Experiments 1~4, we found that when the subliminal prime and supraliminal target were semantically incongruent (e.g. BLUE vs. RED), the incongruency prompted slower target responses. This effect was further shown to be modulated by task load. When the task load was low (Experiments 1 & 3, word naming), the effect appeared in all trials. When the task load was high (Experiments 2 & 4, color naming), the effect appeared only in the later trials when there was a practice effect on the task. This effect was interpreted as a word-induced subliminal semantic interference. However, as the subliminal primes and targets had identical visual forms (BLUE/RED), one may argue that the effect could simply emerge from low-level visual feature adaptation. Although in all experiments, the prime and target were of different font sizes and presented on jittered locations. We first proceeded to examine if prime-target co-localization (on the same or different side of the fixation point) interacted with the word congruency effect with the reasoning that if stronger effect arose when the prime and target were spatially closer, low-level visual adaptation may contribute significantly to our word incongruency effect. We thus ran an additional two-way (word congruency; prime-target location consistency) repeated measures analysis of variance on the data of Experiments 3 & 4. The results from both experiments showed no interaction between location and semantic congruency (Experiment 3, all trials, $F(1, 19) = 0.41$, $p = 0.53$, $\eta_p^2 = .004$; Experiment 4, 4th quarter trials, $F(1, 19) = 0.48$, $p = 0.50$, $\eta_p^2 = .01$). Such results suggested that our findings that prime-target word incongruency slowed down target responses were not merely driven by low-level adaptation of visual stimulus shapes.

However, since our unconscious primes and conscious targets shared not only semantic but also orthographic similarities, it is almost impossible to tease apart if such effect was due to word form (orthographic) incongruency or semantic/conceptual incongruency, or both. To directly examine whether task-induced attention could modulate pure semantic interference, we adopted the design in Experiments 3 and 4 with one adjustment: replacing the target words to “scarlet” and “navy.” By utilizing these two target words, the prime and target now contained no orthographic similarities while still exhibited semantic/conceptual (in)congruencies. It has been shown that once the color association between colors and words is weaker (e.g. naming the color blue on the word *red* vs. *and*), the Stroop interference from color-naming weakens²⁰. The authors also proposed that word frequency could play a key role as low word frequency is typically linked to weaker association with colors. This early observation led to the prediction in our current study that color-naming on the words *navy* and *scarlet* would accompany weaker word semantic interference (i.e. weaker Stroop effect) as demonstrated in the original study. Indeed, in our pilot data ($n = 2$), this

choice of lower-frequency words as targets reversed the relative task load for word naming and color naming: Word-naming was still accompanied by an interference from the color of the target (reverse Stroop effect), while the typical Stroop effect from color-naming disappeared. This opposite pattern of the strength of Stroop/reverse Stroop effects was later confirmed with twenty participants in each experiment. Such pattern teased apart the tight relationship between task load and task set alignment in our previous experiments. That is, the stronger unconscious prime semantic incongruency effect in word-naming experiments can be attributed to either lower task load or task set alignment (attending to semantic feature). While weaker semantic incongruency effect in color-naming can be due to high task load or task set unalignment (attending to non-semantic feature). Thus, it is again almost impossible to pinpoint whether task-induced attentional load or feature-specific attention modulated the unconscious effect. Nonetheless, with the current design, we have created a high-load x task-effect aligned feature-attending (word naming) and low-load x task-effect unaligned feature-attending (color-naming). The new experiments allowed us to examine two separate pairs of conflicting hypotheses on (1) the modulated feature: orthography or semantics and (2) the modulating source: task load or task set. In (1), if pure semantics from an unconscious prime could exert an effect, we expected to see unconscious interference on the target responses. Otherwise if the interference effect was mainly driven by orthographic similarities, the effect would disappear. In (2), if task load played a key role in modulating the interference, the interference would appear in the new low load experiment (Experiment 8, color-naming) but not the new high load experiment (Experiment 7, word-naming, or only in the later practiced trials). Otherwise if task set alignment played a key role, the interference would appear in the word-naming but not the color-naming experiment.

Firstly, we again obtained similar data on successful interocular suppression in both experiments (Experiment 7: location task accuracy 49.81% (1.15%), paired $t(19) = -0.16, p = 0.87$; Experiment 8: 50.2 % (1.33 %), paired $t(19) = 0.15, p = 0.88$). Moreover, the near-ceiling performance was found on the blank and visible trials with 97.78% (0.51%) and 98.12% (1.29%) accuracies in Experiment 7 and 97.81% (0.54%) and 99.69% (0.31%) accuracies in Experiment 8. The critical difference was that the reverse Stroop effect was evident in Experiment 7 (Reaction time comparison of Stroop and non-Stroop trials: $t(19) = 3.15, p = 0.005$, Cohen's $d_{av} = 0.32$) while the Stroop effect was not in Experiment 8 (Reaction time comparison of Stroop and non-Stroop trials: $t(19) = 0.85, p = 0.41$, Cohen's $d_{av} = 0.09$).

In Experiment 7, the RTs in each condition were first normalized against blank trials. A direct paired comparison between semantically congruent and incongruent trials showed null effect ($t(19) = 0.82, p = .42$, Cohen's $d_{av} = 0.19$, Fig. 6A). On average, compared to the blank trials, semantic congruent trials were 2.1 % slower while incongruent ones 0.9 % were slower. We also examined whether prime-target location (same-different) had interacted with the semantic congruency effect. A two-way (semantic congruency; prime-target location consistency) repeated measures analysis of variance was performed on the normalized target word reaction time. Neither the main effect of semantic congruency, $F(1, 19) = 0.09, p = 0.77, \eta_p^2 = .005$, nor the main effect of location congruency was found, $F(1, 19) = 1.32, p = 0.26, \eta_p^2 = .07$. There was no interaction between the two, $F(1, 19) = 0.21, p = 0.65, \eta_p^2 = .01$. Similar results were found in the 4th quarter trials (direct comparison: $t(19) = 1.21, p = .24$, Cohen's $d_{av} = 0.33$; semantic congruency x location consistency 2-way ANOVA: main effect of semantic congruency $F(1, 19) = 0.07, p = 0.80, \eta_p^2 = .004$; main

effect of location: $F(1, 19) = 0.11, p = 0.75, \eta_p^2 = .005$; interaction: $F(1, 19) = 0.49, p = 0.49, \eta_p^2 = .03$.)

In Experiment 8. A direct paired comparison on the normalized RT between semantically congruent and incongruent trials showed null effect ($t(19) = -1.12, p = .28$, Cohen's $d_{av} = 0.24$). On average, compared to the blank trials, semantic congruent trials were 1.32 % slower while incongruent ones 2.7 % were slower. However, a two-way (semantic consistency; prime-target location consistency) repeated measures analysis of variance showed an interaction between the prime-target location and semantic congruency with $F(1, 19) = 8.13, p = 0.01, \eta_p^2 = 0.30$. The main effects were not significant (semantic congruency: $F(1, 19) = 1.1, p = 0.29, \eta_p^2 = 0.06$; location congruency: $F(1, 19) = 0.05, p = 0.83, \eta_p^2 = 0.003$). A planned post hoc comparison between semantic congruent and incongruent trials when the prime-target were co-localized showed a significant effect with $t(19) = -2.45, p = .02$, Cohen's $d_{av} = 0.63$, Fig. 6B).

Fig. 6. Experiments 7 and 8 results. Y-axis denotes reaction time, and X-axis denotes two critical conditions: semantically congruent (SEM-CON) and semantically incongruent (SEM-INCON). Black dots and lines denote longer RT in the semantically incongruent condition while gray dots and lines denote longer RT in the semantically congruent condition. Each pair of dots lined between the two conditions represents one participant. Upward lines indicate longer RT increase in the incongruent condition while downward lines indicated RT decrease in the incongruent condition. The bars denote group mean with the error bars indicating standard error of the mean (SEM). Asterisk denotes significance. Please note that the data from Experiment 8 was from prime-target co-localized trials.

These results showed that, without orthographic similarity, the semantic incongruency between the prime and target did pose a slowing effect on the target responses if that they were co-localized. The observation that the semantic effect further interacted with prime-target co-localization indicated the need of spatial attention toward the primed location for such effect to occur. This semantic effect was also modulated by the task load. In Experiment 8 (color-naming), when the task load was low as indicated by the lack of the Stroop effect, the semantic incongruency slowed down target response. On the contrary, in Experiment 7 (word-naming), when the task load was high as indicated by the reverse Stroop effect, such effect disappeared. As task alignment and task load were separated in these two experiments (i.e. Experiment 7: Semantic task but high load, Experiment 8: Non-semantic task but low-load), distinct from all previous experiments. This particular finding stressed that the level of task load, but not task alignment, modulated an unconscious semantic effect in our paradigm. Table 1 summarizes the stimulus type, task, and effect in all experiments.

Experiment	Prime	Target	Task	Load	Interference
1	BLUE BLUE RED RED		Word-naming	Low	Yes
2	BLUE BLUE RED RED		Color-naming	High	Yes after practice
3	BLUE RED	BLUE BLUE	Word-naming	Low	Yes
4	BLUE RED	RED RED	Color-naming	High	Yes after practice
5	XXXX XXXX		Word-naming	Low	Yes
6	XXXX XXXX		Color-naming	High	No
7	BLUE RED	navy navy	Word-naming	High	No
8	BLUE RED	scarlet scarlet	Color-naming	Low	Yes

Table 1. The table summarizes all eight experiments where we manipulated the color, word, and semantic relationship between the subliminal prime and supraliminal target. Task (word-naming vs. color-naming) and task load (high vs. low, according to the strength of Stroop/Reverse Stroop effect) in each experiment are presented.

Discussion

P.22 Line 8-32. P.23. Line 1-7.

Two important questions were left answered after our first six experiments. Firstly, although the general predictions of the load theory are compatible with our findings, task-induced attention may *selectively* trigger different cognitive sets under different tasks, which in turn gates how an unconscious prime interfered with subsequent target performance. For example, in the word-naming experiments (Experiments 1, 3, & 5), participants' attention was selectively oriented to the word semantics. Such attention tuning would have been applied further to the suppressed prime and allowed semantic interferences to occur. On the contrary, in the color-naming experiments, attention was deployed to the word color, which would have interfered the semantic processing of the target (i.e. the Stroop effect) as well as hindered the semantic interferences from the suppressed prime. Since both our interfering effects were semantic in nature, word- and color-induced semantic congruency effects were gated by this cognitive-set-induced-tuning to different levels: Under word-naming task, both exerted an effect; while under color-naming task, only the more automatic word-induced incongruency effect re-emerged after the participants became more fluent in the task (i.e. 4th quarter trials in Experiments 2 & 4). Secondly, one could argue that since identical word-forms were used as primes and targets, what appeared to be a semantic incongruency effect could be driven merely by orthographic dissimilarity. Although we had jittered the prime-target locations and font sizes, and our further analyses focusing on the interaction between the semantic effect and location yielded null effect, which excluded the possibility of low-level adaptation.

Thus, (1) to ensure that sheer semantic effect could be modulated and (2) to tease apart cognitive-set-induced-tuning and task load modulation, in Experiments 7 and 8, we went one step further to remove the orthographic similarity of the prime and target while maintaining the semantic relationship by introducing a different set of targets: *navy* & *scarlet*. Importantly, the Stroop but not the reverse Stroop effect disappeared when participants were instructed to name the color and word, respectively. Such manipulation thus enabled a misalignment between cognitive-set-induced-tuning attention and task load, allowing us to pinpoint the determining factor in

modulating the unconscious effect. As a result, we showed the unconscious semantic incongruency effect when the task load was low yet non-semantic (Experiment 8, color-naming) but not when the task load was high but semantic (Experiment 7, word-naming), suggesting that general availability of attentional resources, but not cognitive-set-induced-tuning attention, played a key role in modulating the unconscious semantic interference in our study.

4. there is a lot of room for improvements in the writing. There are signs of sloppiness (e.g., repetition from lines 223-232), and many grammatical errors.

Response 3:

We apologize for the writing of the manuscript. We have removed the repetition and revised the manuscript accordingly.

Reviewers' Comments:

Reviewer #1:

Remarks to the Author:

The revised manuscript addressed most of my previous concerns. Specifically, the manuscript is now much more easy to read. In addition, the authors refined their theoretical claims by adding two more experiments. As a result, I can now appreciate the importance of the work.

Nevertheless, I also have a minor concern. In this particular experiment, the authors used a staircase procedure to adjust the luminance level of the prime stimulus and it seems that the luminance level was continuously adjusted throughout the experiment. If that's correct, I have two additional comments.

First, if the luminance level of the prime stimulus was adjusted during the experiment, a 25% detection performance shown in the Supplementary materials is not informative because it should. Instead, it is more common to show the luminance level of the prime. If the luminance levels were comparable across experiments, we can infer that a similar level of suppression was established across experiments. If they were different in a systematic manner, it would be informative how much the load interacted with suppression as well.

Second, for some, using stimulus feature near criterion level by using any adjustment procedure is not satisfactory for arguing that the stimulus is subliminal or invisible even when one uses an interocular suppression paradigm, especially attention was an important factor due to the position uncertainty of the prime.

I acknowledge that the authors emphasized the difficulty of establishing invisibility in the discussion. I also acknowledge that the results remain important if attention and interocular suppression interacted to create some invisibility, where being unconscious is possibly resulted from the preconscious state rather than the subliminal state according to the Dehaene's taxonomy (2006).

Nevertheless, if the adjustment procedure was used during the entire experiment, I would discuss the unavoidable uncertainty a bit more in relation to the adjustment procedure because adjustment procedure which runs with the experiment over different task loads can be more susceptible to the criterion issue, which cannot be easily revealed by any detection task during the experiment.

A more in-depth discussion on this matter can be found in the literature. For me, [Hunt for Artefacts] section of by Kouider & Dehaene (2007) is a guiding principle. Pratte and Rouder (2009) is also informative and relevant in relation to the task load. While both are based on visual masking, there is no reason that interocular suppression should be different when pushing the experimental rigor to the limit. Some technical issues in relation to interocular suppression are discussed in Yang et al. (2014) as well.

Min-Suk Kang

Dehaene, S., Changeux, J.-P., Naccache, L., Sackur, J. & Sergent, C. Conscious, preconscious, and subliminal processing: a testable taxonomy. *Trends Cogn. Sci.* 10, 204–211 (2006).

Kouider, S. & Dehaene, S. Levels of processing during non-conscious perception: a critical review of visual masking. *Philos. Trans. R. Soc. London Ser. B, Biol. Sci.* 362, 857–875 (2007).

Pratte, M. S. & Rouder, J. N. A task-difficulty artifact in subliminal priming. *Atten. Percept. Psychophys.* 71, 1276–1283 (2009).

Yang, E., Brascamp, J., Kang, M.-S. & Blake, R. On the use of continuous flash suppression for the

study of visual processing outside of awareness. *Front. Psychol.* 5, 1–17 (2014).

Reviewer #3:

Remarks to the Author:

I appreciate the efforts put in by the authors in revising their manuscript.

Regarding task load vs. task set, the authors stated: "We took advantage of such asymmetry of task load in responding to different aspects of the stimulus and defined color-naming as high-load and word-naming as low-load in our study". But see the study from MacDonald AW et al. (*Science* 2000), in which task preparation (color vs. word task) and performance monitoring (congruent vs. incongruent) were explicitly dissociated, using a Stroop task such as the one used here. During the time before the target was presented, the difference was in task set.

The two new experiments (7 & 8) are helpful, but they also add more complexity into the interpretation. The new experiments were aimed at dissociating task set alignment and task load. Here the "task load" (high or low) was based on the strength of Stroop and Reverse Stroop effect. I think this is a rather indirect measure of the task load, I would prefer to see the original (unnormalized) RT data to get a more direct sense of the load. In addition, the Word and Color conditions differ in more ways than just the task load in these two new experiments.

In any case, the authors misunderstood my original comment #3. I was pointing out that the 'gating' effect from task or load could occur at the 'feature/semantic information' extraction stage during priming, or it could occur at the 'interference' stage during test. The authors' response was about the distinction between task set vs. task load, but did not address where the 'gating' was applied. For example, there is a reasonable chance that the information extraction from the invisible primes (well before the target presentation) was indifferent to the task load, but the potency of interference from the prime to the target was modulated during the time when the visible target was presented. The bottom line is that during the 4 sec presentation of the prime, participants were not performing the high load or low load task, the task was subsequent to the prime.

The writing is still quite difficult to follow.

Reviewers' comments:

Reviewer #1 (Remarks to the Author):

The revised manuscript addressed most of my previous concerns. Specifically, the manuscript is now much more easy to read. In addition, the authors refined their theoretical claims by adding two more experiments. As a result, I can now appreciate the importance of the work.

Nevertheless, I also have a minor concern. In this particular experiment, the authors used a staircase procedure to adjust the luminance level of the prime stimulus and it seems that the luminance level was continuously adjusted throughout the experiment. If that's correct, I have two additional comments.

First, if the luminance level of the prime stimulus was adjusted during the experiment, a 25% detection performance shown in the Supplementary materials is not informative because it should. Instead, it is more common to show the luminance level of the prime. If the luminance levels were comparable across experiments, we can infer that a similar level of suppression was established across experiments. If they were different in a systematic manner, it would be informative how much the load interacted with suppression as well.

First of all, we thank the reviewer for the previous comments, which helped improve the manuscript. Please see our replies below.

Response 1:

As the reviewer had correctly inferred, the luminance level of the prime stimulus was continuously adjusted to ensure proper suppression. The reviewer is absolutely correct that the 25% detection performance was expected given that participants exhibited **proper** breaking suppression behavior. For example, if some participants were reluctant to report visibility when they had already detected the prime, these participants' detection performance would have deviated from 25%.

As Experiments 3, 4 and 7, 8 used the colorless (gray) primes and identical trial-by-trial thresholding procedures, we looked into the breaking thresholds of these experiments to see if they were comparable as the reviewer suggested. The average prime luminance levels in Experiments 3, 4, 7 and 8 were 10.58 (6.7) %, 9.4 (5.5) %, 11.12 (1.04) %, and 10.4 (0.84) %, with no difference between the four ($F(3, 76) = 0.81, p = 0.49$). However, we have to point out that in our study, different participants were recruited in different experiments, thus the luminance levels across experiments may not be a good indicator of suppression strength. For instance, different individuals might have intrinsic differences on how much luminance was required to break suppression.

Revision:

Supplementary information

2. Invisibility of the suppressed prime

Furthermore, as we have used a floating thresholding procedure in all experiments, one may argue that different levels of subliminal prime luminance could occur and cause the differences across experiments. As Experiments 3, 4 and 7, 8 used the colorless (gray) primes and identical trial-by-trial thresholding, we looked into the breaking thresholds of these experiments to see if they were comparable. The average prime luminance levels in Experiments 3, 4, 7 and 8 were 10.58 (6.7) %, 9.4 (5.5) %, 11.12 (1.04) %, and 10.4 (0.84) %, with no difference between the four, $F(3, 76) = 0.81, p = 0.49$. However, we have to point out that in our study, different participants were recruited in different experiments, thus the luminance levels across experiments might not be a good indicator of suppression strength.

Second, for some, using stimulus feature near criterion level by using any adjustment procedure is not satisfactory for arguing that the stimulus is subliminal or invisible even when one uses an interocular suppression paradigm, especially attention was an important factor due to the position uncertainty of the prime.

I acknowledge that the authors emphasized the difficulty of establishing invisibility in the discussion. I also acknowledge that the results remain important if attention and interocular suppression interacted to create some invisibility, where being unconscious is possibly resulted from the preconscious state rather than the subliminal state according to the Dehaene's taxonomy (2006).

Nevertheless, if the adjustment procedure was used during the entire experiment, I would discuss the unavoidable uncertainty a bit more in relation to the adjustment procedure because adjustment procedure which runs with the experiment over different task loads can be more susceptible to the criterion issue, which cannot be easily revealed by any detection task during the experiment.

Response 2:

We entirely agree with the reviewer that simply using the near threshold presentation to establish prime invisibility was not enough. However, in our study the prime invisibility was not simply established by a single criterion but instead by the combination of all the following factors: immediate subjective report, retrospective 2AFC chance rate prime localization, participants' proper detection behavior (near ceiling performance on the visible catch trials, low false breaking rate, as well as stable breaking threshold). Furthermore, our additional analysis on those participants that passed the chance-level prime localization (binomial test, $p > .05$) showed similar results. A combination of all factors is how we claim that the prime was indeed invisible to the participants. For a detailed explanation, please see supplementary information "Invisibility of the suppressed prime."

However, we do agree that even when the prime was indeed invisible to the participants. Task load and the floating thresholding procedure may together affect how invisibility was achieved, although this was not seen in the prime luminance differences as we reported in our response 1.

A more in-depth discussion on this matter can be found in the literature. For me, [Hunt for Artefacts] section of by Kouider & Dehaene (2007) is a guiding principle. Platte and Rouder (2009) is also informative and relevant in relation to the task load. While both are based on visual making, there is no reason that interocular suppression should be different when pushing the experimental rigor to the limit. Some technical issues in relation to interocular suppression are discussed in Yang et al. (2014) as well.

Response 3:

We agree that on the issue of ensuring proper invisibility, there should not be any difference between interocular suppression, masking, or any other techniques. That was why we had embedded in multiple experimental and statistical measures into our study to achieve and examine prime invisibility. For instance, as the reviewer pointed out, under the section of *The Hunt for Artefacts*, Kouider and Dehaene (2007) reviewed several drawbacks that previous subliminal semantic priming paradigms suffered from. For instance, the prime visibility was sometimes underestimated because the invisibility was defined by a *separate* thresholding procedure where participants exhibited chance performance on the visibility measure. Due to the differences between the thresholding phase and the experimental phase, participants prime identification could go up to way above chance as indicated by a later study. Also, some studies suggest that prime visibility correlated with the semantic effects, thus suggesting that having visibility of the prime may be crucial for such subliminal effects.

Please note that these drawbacks were carefully prevented in the current study as we adjusted the luminance continuously to avoid underestimation of prime visibility. That is, if other factors were to modulate invisibility of the prime, such as prime-target relatedness, or task load, these additional factors would be taken care of by this dynamic thresholding procedure. Moreover, we did not find any luminance differences across experiments, showing that even if other factors could have intervened the thresholding procedure, the effects were minimal in our study. Here we also examined the correlation between the prime detection performance and the unconscious semantic effect and showed that there was no correlation between the two in our main experiments (Exp 3 word-naming all trials: $r = -0.29$, $p = 0.21$; Exp 4 color-naming, 4th quarter trials: $r = -0.03$, $p = 0.91$; Exp 8 color naming all trials: $r = 0.21$, $p = 0.37$).

Particularly on the weakness of interocular suppression, as Yang et al. (2014) have pointed out, there is a “gray zone” in the paradigm where participants may experience partial visibility, which contaminates an unconscious effect. Furthermore, a prolonged duration of detection failure may lead to an underestimation of the observer’s detection performance due to the lack of attention or fatigue. These were also the most important issues that we intended to address in our experiments. For instance, a continuous thresholding procedure could ensure that participants engage in the prime detection task. Otherwise if the participants failed the detection task, the thresholding results would have been a failure. And the partial awareness possibility was carefully addressed through multiple controls (please see supplementary information “Invisibility of the suppressed prime”).

As the reviewer pointed out, Platte and Rouder (2009) found an effect of prime task difficulty on subliminal priming effects, and in the context of the findings of the time, that was interpreted as evidence of a visibility artifact, because they did not anticipate that task difficulty would be a factor in the downstream processing of invisible primes. Since then, there has been increasing evidence

of attention load and attention set on various forms of subliminal visual processing (e.g. Kiefer & Martens, 2010; Hsieh, Colas, & Kanwisher, 2011). Our results replicate Platte and Roulder (2009), but by controlling additional factors and leaning on the more recent findings regarding attentional load and attentional set, we are able to advance an alternative proposal - that all of these results support the role of task difficulty in modulating high-level semantic processing of invisible primes. However, why in their study correct prime classification (i.e. some level of prime visibility as the authors suggested) was crucial to the priming effect? We have to point out that there are several differences between their and our paradigms. For example, the visual suppression techniques were different. Under our interocular suppression, the prime was unconsciously presented for > 1.5 seconds, and thus potentially strengthened the raw stimulus power. Moreover, the nature of the tasks was largely different, as the authors utilized the classical paradigm from Dehaene et al. (1998) to see if number categorization on a target could be affected by a preceding prime number, our study directly used words to examine unconscious word semantic processing. Intrinsic stimulus differences might lead to very different findings (e.g. stimulus automaticity).

Min-Suk Kang

Dehaene, S., Changeux, J.-P., Naccache, L., Sackur, J. & Sergent, C. Conscious, preconscious, and subliminal processing: a testable taxonomy. *Trends Cogn. Sci.* 10, 204–211 (2006).

Kouider, S. & Dehaene, S. Levels of processing during non-conscious perception: a critical review of visual masking. *Philos. Trans. R. Soc. London Ser. B, Biol. Sci.* 362, 857–875 (2007).

Platte, M. S. & Roulder, J. N. A task-difficulty artifact in subliminal priming. *Atten. Percept. Psychophys.* 71, 1276–1283 (2009).

Yang, E., Brascamp, J., Kang, M.-S. & Blake, R. On the use of continuous flash suppression for the study of visual processing outside of awareness. *Front. Psychol.* 5, 1–17 (2014).

Reviewer #3 (Remarks to the Author):

I appreciate the efforts put in by the authors in revising their manuscript.

Regarding task load vs. task set, the authors stated: "We took advantage of such asymmetry of task load in responding to different aspects of the stimulus and defined color-naming as high-load and word-naming as low-load in our study". But see the study from MacDonald AW et al. (Science 2000), in which task preparation (color vs. word task) and performance monitoring (congruent vs. incongruent) were explicitly dissociated, using a Stroop task such as the one used here. During the time before the target was presented, the difference was in task set.

The two new experiments (7 & 8) are helpful, but they also add more complexity into the interpretation. The new experiments were aimed at dissociating task set alignment and task load. Here the "task load" (high or low) was based on the strength of Stroop and Reverse Stroop effect. I think this is a rather indirect measure of the task load, I would prefer to see the original (unnormalized) RT data to get a more direct sense of the load. In addition, the Word and Color conditions differ in more ways than just the task load in these two new experiments.

We thank the reviewer for helping us improve our manuscript. Please see below our replies.

Response 1:

We thank the reviewer for pointing out this important study. It is possible that the task load differences generated by our color/word-naming tasks were a mixture of task preparation and performance monitoring. We do need to point out that unlike Macdonald et al. (2000), we did not implement a task-switching scheme in our study. Indeed, in the color-naming and word-naming experiments, participants could have had distinct task preparations for the upcoming target stimulus that interacted with how a subliminal stimulus was processed.

We agree with the reviewer that using the strength of Stroop/reverse Stroop effects to define task load was indirect. The raw average RTs in Experiments 7 (word naming, reverse Stroop effect) and 8 (color-naming, Stroop effect) were 1036 ms vs. 974 ms ($t(38) = 0.9, p = 0.37$). Please note that different participants were recruited in these two experiments and the comparison of raw RTs may not be a good direct indicator of task load across experiments.

As the reviewer correctly pointed out, the Word and Color conditions at least differed in another significant way: task set, which was what Experiments 7 & 8 were designed to tease apart and examine.

In any case, the authors misunderstood my original comment #3. I was pointing out that the 'gating' effect from task or load could occur at the 'feature/semantic information' extraction stage during priming, or it could occur at the 'interference' stage during test. The authors' response was about the distinction between task set vs. task load, but did not address where the 'gating' was applied. For example, there is a reasonable chance that the information extraction from the invisible primes (well before the target presentation) was indifferent to the task load, but the potency of interference from the prime to the target was modulated during the time when the visible target was presented. The bottom line is that during the 4 sec presentation of the prime, participants were not performing the high load or low load task, the task was subsequent to the prime.

The writing is still quite difficult to follow.

Response 2:

We apologize for misinterpreting the reviewer's original question. We have now added in a discussion where possible gating mechanisms were explicitly spelled out.

Revision:

Discussion

P.23. Line 29-32.

P.24. Line 1-7.

...However, we have to point out that where in the perceptual/neural pathway exactly did such task-induced attention gating occurred remained elusive. For instance, high task load could impose a higher need for attentional resources for the conscious stimulus and thus leave lesser attentional resources for the unconscious stimulus in a general, blocked manner. That is, the effect of task

load was effective on the whole experimental block rather than at the single trial level. It is plausible that the gating occurred during the presentation of the unconscious stimulus, even prior to the target presentation. On the other hand, the gating could have occurred at a much later stage during the interference between the unconscious and conscious stimuli. That is, the extent to which the prime stimulus was processed was similar under different task loads, however, the task load modulation occurred at the interface of prime and target, when the target was shown. Future experiments are required to examine these possibilities.

Reviewers' Comments:

Reviewer #1:

Remarks to the Author:

All my previous concerns were addressed.

Reviewer #3:

Remarks to the Author:

With the additional clarification regarding where the gating effect could be applied added in the discussion section, the authors have addressed my concerns.